# A multi-level multi-product supply chain network design of vegetables products considering costs of quality: A case study

Sareh Khazaeli[1]*, Ramazan Kalvandi[2], Hadi Sahebi[1]

1 Industrial Engineering, Iran University of Science and Technology, Narmak, Tehran, IR, 2 Agricultural Garden, Yaman Avenue, Shahid Chamran Highway, Tehran, IR

* khazaeli_sareh@ind.iust.ac.ir

**Data Availability Statement:** The most critical data are presented in Supporting Information files. All are not presented due to the high space they need. If there is no space limitation in the paper, it can be published.

## Abstract

Effective logistics management is crucial for the distribution of perishable agricultural products to ensure they reach customers in high-quality condition. This research examines an integrated, multi-echelon supply chain for perishable agricultural goods. The supply chain consists of four stages: supply, processing, storage, and customers. This study investigates the quality-related costs associated with product perishability to maximize supply chain profitability. Key factors considered include the network design, location of processing and distribution centers, the ability to process raw products to minimize post-harvest quality degradation, the option to sell the excess produce to a secondary market due to unpredictable yields, and the decision not to fulfill demand from distant customers where significant quality loss and price drops would be involved, instead diverting those products to the aforementioned secondary market. Quantitative methods and linear mathematical programming are employed to model and validate the proposed supply chain using actual data from a real-world case study on vegetable supply chains. The main contribution of this research is the incorporation of quality costs into the objective function, which allows the supply chain to prioritize meeting nearby customers' demands with minimal quality loss over serving distant customers where high quality loss is unavoidable. Additionally, deploying a faster transportation fleet can significantly improve the overall profitability of the perishable product supply chain.

## 1. Introduction

Vegetables are perishable, edible, agricultural products that deteriorate during a limited shelf life [1]. Quality of perishable products is essential to the customer because such products deteriorate fast and endanger the consumer's health [2]. There is a consensus in the literature on the reasons why people buy organic food; however, there is also a gap between the consumers' generally positive attitude toward organic food and their relatively low level of actual purchases [3]. Quality of vegetables is one of the important measures to its customers due to the quality deterioration rate of products which relates to the health of consumers [2]. Time decay and shortages are common phenomena in products with short life cycles, and financial volatility necessitates a more accurate characterization of inventory costs based on time-adjusted value

**Funding:** The author(s) received no specific funding for this work.

[4]. The supply chain management concept evolved when manufacturers experienced a strategic partnership with their direct suppliers. Then the logistics and transportation experts improved it one step forward and involved the distribution and transportation operations. Next, the concept of integrated logistics was recognized as the supply chain management [5]. Product quality is another novel concept in the supply chain management [6]. Moreover the quality deterioration often happens in traditional supply chains which, for the most part, are poorly planned [7]. From a product quality perspective, when processed products decay at a faster rate than raw materials, storing raw materials is favored [8]. Alternatively, when processing decreases the quality decay rate, a short time until processing is favored [9]. The supply chain (SC) of vegetables consists of four echelons: 1) purchasing raw materials, 2) processing, 3) distribution, and 4) customers to which products are delivered [10]. Since perishable products (agri-foods) have limited shelf life, logistic-related topics are important in business [11]. Transportation share in supply chain costs reached about 92% in the distribution sector in some traditional chains [12]. The post-harvest pre-customer-sent product loss [13] accounts for more than 40% of the supply chain costs even in industrialized and developed countries [14]. It occurs in terms of both the product quantity and agri-food quality loss throughout the chain and imposes quality costs on the chain [15]. Although considering the shelf life losses is in relation to an increase in transportation costs, it worth investing on transportation infrastructure due to less quality loss. Moreover, from a system's point of view, integrating warehousing and transportation in the supply chain can highly affect the total cost, customer satisfaction and inventory level. Integrated models of providing and storing perishable products help to maximize meeting demands [11]. Integration of storing and distributing decisions leads to more efficiency than other operational integration [16, 17]. Integration of strategic decision making and operational processes appears relevant, especially for such perishable products as agri-foods [18]. Recently some strategies were studied in supply chain management of perishable products to control the perishability of products which are inventory management [19], reverse logistic management [20], pricing [7], and robust optimization [21].

Notably, product quality is characterized by the product's remaining shelf life and thus is time-dependent [22]. Taguchi described the deviation in performance using the quality loss function that measures the product's quality loss in terms of the total loss to society due to functional variation and harmful side effects [23]. For perishable foods, product quality degradation must be identified because it significantly affects consumers' decisions and retailer profitability [22]. On the other hand, computing the cost of quality loss for an integrated supply chain allows for exploring the interrelationships among business entities. It enables the supply chain to achieve a minimum total cost by investing in quality and, hence, increasing the overall benefit [24]. Today, lateral marketing is the most effective way of competing in mature/immature markets, where micro-segmentation and plenty of brands don't leave any space for new opportunities [25]. One of problems in the perishable agricultural products' supply chain is a high quality loss post-harvest, which leads to different quality costs and the customer dissatisfaction. A brief review of the literature reveals that rarely is there any established advanced multi-echelon vegetable supply chain wherein the profit is maximized by considering such features as product quality degradation, quality loss-related costs, and settling lateral markets. Due to this research gap, current study is aimed to maximize the profit of perishable products supply chain considering their related quality costs. The question in this research is how considering both the cost of qualities and the second market in the supply chain network design (SCND) of perishable products can affect the benefits of stakeholders, such as farmers and customers in the supply chain.

The research objective is to formulate a SCND of perishable products by considering different costs of qualities in the supply chain and settling a lateral market and processing the part of

perishable products that have not entered the supply chain due to its high level of perishability and enters to the second market be used in specific form satisfying customers, in the mathematical mixed integer linear programming. The current study intends to make affecting decisions in different levels of decision making as: 1) strategic level; locating different centers in the supply chain, 2) tactical level; determining the processing type, and quantities of different products be delivered to the customers, and 3) operational level; selecting a suitable mode of transportation and quantities in the SCND. To address this challenging problem, vegetables, important perishable products, were examined in a case study by first studying the multi-echelon agri-food supply chain (AFSC) based on the post-harvest quality features.

The remainder of this paper is structured as follows: In the next section, a brief overview of related literature reviews on the quality management of perishable agricultural products is given. Section 3 describes the research methodology, a quantitative supply chain modeling approach in a linear programming framework. The case study and sensitivity analysis results in the optimum point are presented in Section 4, the research conclusions in Section 5, managerial implications in Section 6, and future research and limitations in Section 7.

## 2. Literature review

### 2.1. Agricultural products supply chain

Customers pay special attention to the quality and safety of agri-foods because they directly affect their health [26]. This quality can be measured by such different criteria as the purchasability [27], lifetime (day) left [28], color [29], freshness [30] and light-greenness of vegetables (L. in the Hunter Laboratory) [31, 32]. Creating an efficiency-responsiveness balance in quality-based customer-oriented supply chains is worth considering [9]. The optimal operation strategy is acquired based on product quality [6]. Organizations that have instituted a system of quality cost measures have experienced dramatic positive results because it translates the implications of poor quality, activities of a quality program, and quality improvement efforts into a monetary language for managers to understand which factors are important in affecting profitability and the consumer need [24].

Decisions made in the supply chain of perishable products are strategic, tactical and, operational; strategic decisions that have long-term effects on firms are those made on the network design, supply chain network design [33] and the location of different equipment in the processing, distribution and, hub centers to make the best use of the capacity of the existing facilities [34]. In the strategic level of decision-making in the perishable products' supply chain design, different ways to cope with increasing product quality decay can be identified. On the one hand, the network can be centralized to decrease handling time (for each transport to a hub, a fixed handling time is incorporated in the transport time) and hence decay. On the other hand, more hubs can be opened to decrease transport time and decay [9]. Moreover, technical models are popular and have public applications in harvest programming, product selection, and labor capacity in agricultural products supply chains. Besides strategic and tactical decisions, the supply chain also involves operational decisions for which it is assumed that the former two are already known and sufficient knowledge is available about production, demand, and transportation [35]. Pasha et al. studied an integrated bi-objective quality-based production-distribution agri-food MILP supply chain model in which profitability is maximized by defining the quality as a function of such decisions as the location of hubs and transportation strategy throughout the supply chain [17], whereas making decisions in an integrated way will reduce costs compared to individual decisions made at each level [36, 37]. Moreover, in the greenery supply chains, De Keizer et al. presented a model in which decisions made on the greenhouse location (strategic) are based on the plant's lifetime in that location

[9]. As changes in the temperature and enthalpy levels change the food quality [38], Khazaeli et al. and Rong et. al determined the temperature of distribution centers and deliveries to meet the expectations of different customers as the operational decision-making in a supply chain management [39, 40].

## 2.2. Quality of agricultural products

In most supply chain designs, cost, profit, quality, responsiveness and environment are the general decision-making factors [34]. Although cost and profit are still the main criteria in almost all quantitative mathematical programming models of the supply chain of perishable agricultural products, in recent years, other criteria, such as product quality [9, 17, 18, 41, 42] and environmental protection [43] have also been considered in some studies. The quality function of perishable agricultural products can be either complex or simple [44]. It has been shown that, the decrease of a single quality attribute of agricultural products can be approximated by one of the four basic types of mechanism which are zero-order reactions having linear kinetics, Michaelis Menten kinetics, first-order reactions having exponential kinetics, and autocatalytic reactions with logistic kinetics [45, 46]. For the concept of keeping quality, it is convenient to assume zero-order reaction kinetics [28], and mostly the Michaelis Menten kinetics reduces to a linear one in the initial region of decay, which is the most important in quality assessment [47]. Therefore, the quality variable of vegetables in the initial region of decay can be considered in a widely used equation, in which the quality function changes by the time linearly. It is shown in Eq 1.

$$\frac{dQ}{dt} = kQ(t) = Q_0 - k.t \qquad (1)$$

Where, $Q_0$ is the initial quality, $t$ is time and $k$ is a degradation rate. In a dynamic environment, the well-known Arrhenius equation shows that the degradation rate ($k$) depends on the activation energy of the material, and the environmental factors [28, 48, 49].

The perishable products' quality model shown in Eq 1 has been frequently used to capture the degradation of food products over time. For example, in the grocery retail chain, Wang and Li presented a pricing model to maximize food retailer's profit in a dynamically identified food shelf life by using Eq 1 [50]. Chen and Chen proposed an on-site direct-sale dynamic supply chain inventory model, considering time-dependent quality losses for perishable foods [22]. Lejarza and Baldea presented a closed-loop, feedback-based control framework, that employs real-time product quality measurements for optimal supply chain management [51]. Moreover, Xu et al. presented a real time decision support framework to mitigate the quality degradation in the journey of agricultural perishable products from farm to the retailer in the supply chain based on the Eq 1 [52].

Generally, cost, benefit, and quality factors are the most important factors that are to be optimized in network designs. Mostly, agri-food should make a logical balance between two topics, which are the price reduction and the customer service improvement [38]. In the field of multi-objective supply chain network design, De Keizer et al. and Khazaeli et al. showed that, the quality of agricultural products causes cost in the supply chain's network [18, 39]. A review of quantitative supply chain research on the perishability of agri-food by considering related quality costs is summarized in Table 1.

## 2.3. Research gaps and contributions

Due to the importance and necessity of developing SCM from a larger perspective to provide a win-win situation for each participant in the supply chain, in this paper, we aim to develop a novel mathematical model to design a supply chain network, based on quality function

**Table 1. Summary of mathematical SC models based on the quality of perishable products.**

| Author/Year → Feature of the model | | [7] | [39] | [51] | [17] | [22] | [12] | [34] | [53] | [52] | [54] | [9] | [27] | [43] | [41] | [18] | [42] | [35] | [40] | Current research |
|---|---|---|---|---|---|---|---|---|---|---|---|---|---|---|---|---|---|---|---|---|
| **Mathematical models** | Optimization- (LP) | | | | | * | | | | | | | | | | | | | | |
| | Optimization- (MILP) | | | | | | * | * | * | | | * | * | * | | * | | * | | * |
| | Optimization- (NLP) | | | * | * | | | | | * | | | | | * | | | | | |
| | Optimization- (MINLP) | * | * | | * | | | | | | * | | | | | | | | | |
| | General mathematical models | | | | | | | | | | | | | | | | * | | | |
| **Dynamic or static** | Static | | | | | | | | | | * | * | * | | | * | | | | |
| | Dynamic | * | * | * | * | * | * | * | * | * | | | | | * | | * | * | * | * |
| **Flow direct** | Forward | * | * | | * | * | * | * | * | * | * | * | * | * | * | * | * | * | * | * |
| | Backward | | | * | | | | | | | | | | * | * | | | | | |
| **Uncertainty** | Certain | * | * | * | * | * | | * | | | * | * | * | * | * | * | | | * | * |
| | Stochastic | | | | | | | | | | | | | | | * | | | | |
| | Robust | | | | * | | | * | | * | | | | | | * | * | | | |
| **Decision level** | Strategic with location | * | * | * | | * | | * | * | | | * | * | | | | | | | * |
| | Strategic without location | | | | | | | | | | * | | | | | | | | | |
| | Tactical | * | * | * | | * | | * | * | * | | * | | * | | * | * | * | | * |
| | Operational | * | * | | * | | * | | | | | * | | * | * | | | * | * | * |
| **Objectives of programming** | Economical | * | | * | * | | * | * | * | * | * | * | * | * | * | * | * | | * | * |
| | Environmental | * | | | | | | * | | | | | * | | | | | | | |
| | Social | * | | | | | | | | | | | | | | | | | | |
| | Quality- based | | * | | * | * | | * | * | | | * | * | * | | * | * | | | * |
| **No. of product** | One- product | | | * | | * | | | | | * | | * | * | | * | | * | | |
| | Multi product | * | * | | * | | | * | * | * | | * | | | * | | * | | | * |
| **Network element** | Material supply | * | * | * | * | * | * | * | * | * | * | * | | * | * | * | * | * | | * |
| | Storage | | | | | | | | * | * | * | * | * | | | | * | * | | * |
| | Process | * | * | | | | | * | * | * | | * | | | | * | * | | | * |
| | Distribution | * | * | | * | * | | * | * | * | * | * | * | * | | * | * | * | * | * |
| | Retailer/ Customer | * | * | * | | * | * | * | * | * | * | * | | * | | * | | | | * |
| | Transportation | * | * | * | | * | | * | | | * | * | | * | * | * | | * | | * |
| Number of echelons of the supply chain | | 4 | 4 | 2 | 2 | 2 | 3 | 3 | 4 | 5 | 4 | 4 | 1 | 3 | 2 | 3 | 2 | 5 | 3 | 4 |
| **Study field** | Livestock/agriculture food | * | | | * | | | * | | | * | * | * | * | | | | * | | |
| | Vegetables/ flowers/ Herbal plants | | | * | * | * | | * | | * | | | | | * | * | * | | | * |
| | Pharmacy and herbal medicines | | * | | | | | | | | | | | | | | | | | |

**Source(s)**: Authors' work

elements in the vegetables' sector. The summary of the literature review outlines the gaps in the literature as follows:

1. Despite the importance of the cost of qualities in designing supply chains due to the perishability of the products, the cost of quality concept has not been widely incorporated by researchers in the design of agricultural products' supply chains.

2. No research has paid attention to the lateral market to look at the quality problems from the side covering some target customers.

3. Few researchers have considered the benefits of several stakeholders of the agricultural supply chains simultaneously. The stakeholders in agricultural Products' supply chain are consumers, farmers, the environment, and society.

The proposed SCND is a multi-product, multi-echelon model with exact (certain) demand that makes decisions at strategic, tactical, and operational levels. It has focused on "quality" by considering the quality deterioration which is time-dependent in the initial region of decay, moreover, by defining costs of quality degradation in the quality-cost functions. Features that differentiate the present research from others are displayed in the last row in Table 1. As previous researches have demonstrated, traditional supply chain of agri-food is unstructured, which generally leads to low quality and low benefit of agricultural products, the presented research is developed, in which the main contributions are as follows:

✓ Providing a network design model for an integrated multi-level supply chain of perishable products wherein profit is optimized by considering quality decay aspect of the products.

✓ Optimizing the profit of the supply chain of perishable products considering different quality costs for them due to unmet demand, product waste and reduced revenue of low-quality products.

✓ Introducing a strategy of selling perishable products to lateral markets before letting products enter the chain to prevent the production of low-quality products along it.

✓ Enabling the purchase of the farmer's total agricultural product above the contract ceiling due to unpredictable production to prevent waste production and its scattering in the environment.

✓ Introducing a strategy of producing semi-processed, low-quality products (from those that did not enter the chain) to meet part of the market demand for lower-quality lower-price products.

The developed model is a four-echelon supply chain of perishable agricultural products in which the time-dependent quality of the products is considered. In addition, a lateral market is considered in the designed supply chain that does not stand higher than vertical marketing and completes the primary market.

In the end, the developed model is applied to a case study of a firm in the agricultural products industry with four echelons of farm-processing-distribution-customer centers. The vegetables selected as candidates for the present supply chain network design are *Yarrow*, *Borage flower*, and *Melisa*, due to their priority in agricultural studies and their application in various industries [55].

Although there are some studies done to minimize quality losses of perishable products by multi-objective problem-solving approaches [17, 19, 20, 21, 39], the programming in the present research is done as a single objective problem solving by profit objective function underlying quality loss costs.

## 3. Problem description and formulation

From the perspective of the research approach, this research is quantitative, done as a mathematical mixed integer linear programming (MILP) modeling with the objective function of profit by considering the cost of quality factors of products in the multi-echelon perishable products' supply chain. It is applicable to the related supply chains. It focuses on an integrated

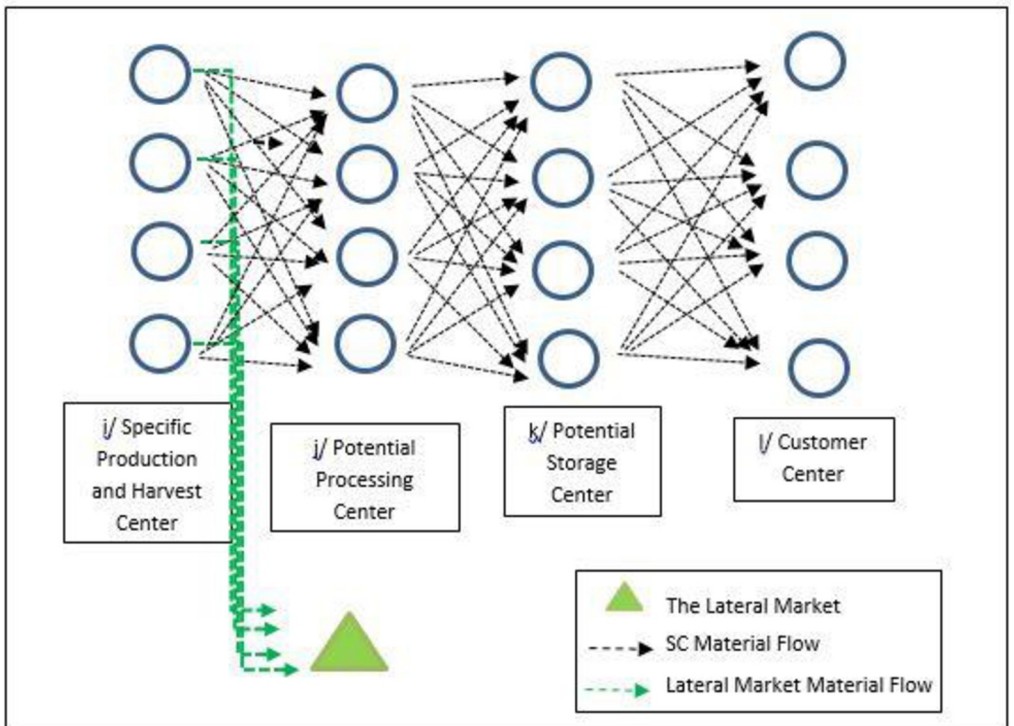

**Fig 1. Flow diagram of the agricultural products' SC.**

multi-product SCND of agricultural products that provides, processes, stores and distributes materials. It considers customer demands and sells the farmers' in-excess products to the second market. The designed model was solved using GAMS 24.1.2 software by exact solution method by epsilon-constraint. The model is validated by applying it in the case study of a multi-vegetable supply chain of a firm in a fertile area in Iran country. The designed supply chain of the firm is shown in Fig 1.

First, products through related contracts and in-excess products are bought from farmers in the study area. In the second echelon of the proposed supply chain, some or all of the purchased products are processed at related centers resulting in different degrees of product quality. Third, the products in the former echelon are stored in cool storage centers until being distributed and fourth, they are sold to wholesalers. Another part of the purchased products are transferred to the second market as lower quality products in different industries (tea bags, spices in food, etc.). Different road modes of transport are used between different echelons of the supply chain.

The modeling makes decisions at different echelons of the supply chain. Decisions made are (1) selecting farms and the quantity of raw products to be purchased from each of them, (2) the quantity of products sold to the second market, (3) the number of processing and storage facilities to be settled in the supply chain, (4) product flow and the vehicles to carry out the transportation between the active facilities i. e. from farms to wholesalers and (5) assignment of processing facilities to the products. They are made based on minimizing the total cost of the supply chain design considering the cost of qualities. In the following, assumptions and the modeling are described.

### 3.1. Assumptions

- The location of production centers is specified.

- Capacities of the processing centers, and also storage centers are determined.

- Customer demand for each type of processed product is pre-determined.

- Shortage to customer demand is allowed.

- The quality of products post-harvest in the supply chain is considered time-dependent.

- The deterioration rate of each product is considered specific, based on the activation energy of the material.

- The approach of quality costs is considered in measuring the quality of products in the objective function modeling.

- The transportation speed of each mode is assumed uniform.

- In-excess products are sold to the second market.

- Over-time quality loss-related cost, unmet customer demand and product waste are considered as quality costs.

- The cost of the lost product quality equals the price drop in proportion to the quality drop by a factor of ten (The coefficient (10) is proposed by experts based on pairwise comparisons of cost and quality criteria).

- The quality cost of the customer credit for each demand equals the revenue lost due to not meeting one unit demand.

- The quality cost of the product waste equals the revenue from the product sales not realized, causing that product to enter the environment as waste.

- Products are bought from farmers: 1) at a price for first-grade products based on the amount in the contract and 2) at a price for second-grade products for those over that in the contract (According to experts, the purchase-price-drop coefficient is 0.3 in the market).

- Products are sold to the supply chain customers at a price for first-grade products and those outside the supply chain are sold in the second market at a price for second-grade products (According to experts, the sell-price-drop coefficient is 0.3 in the market).

The mathematical model, its objective and its constraints are presented in the following.

## 3.2. Mathematical modelling

Symptoms used in the model consist of sets, related indexes, parameters and variables, objective functions and constraints, are as follows:

### Sets and indexes.

| | |
|---|---|
| $i \in I$ | *Set of contracted farms* |
| $j \in J$ | *Set of potential processing centers* |
| $k \in K$ | *Set of potential storage centers* |
| $l \in L$ | *Set of customer centers* |
| $n \in N$ | *Set of harvested products* |
| $p \in P$ | *Set of processing technologies* |
| $m \in M$ | *Set of transportation modes* |

**Parameters.**

| | |
|---|---|
| $pd_i^n$ | *Amount of product n produced at location i (ton)* |
| $pri_n$ | *Price of product n (USD/ ton)* |
| $pri'_n$ | *Price of purchasing second-degree product n from farmers (USD/ ton)* |
| $Sa_n^p$ | *The sale price of processed product n in type-p form (USD/ ton)* |
| $sa'_n$ | *Price of selling second-degree product n to the second market (USD/ ton)* |
| $cap_{jp}$ | *Processing center j capacity for the type-p processed product (ton)* |
| $cap_{kP}$ | *Storage center k capacity for the type-p processed product (ton)* |
| $cap_{mp}$ | *Transportation mode m capacity for type-p processed product (ton)* |
| $d_{ij}$ | *Distance between farm i and processing center j by Google (km)* |
| $d_{jk}$ | *Distance between processing center j and storage center k by Google map (km)* |
| $d_{kl}$ | *Distance between storage center k and wholesaler l by Google map (km)* |
| $V_m$ | *The average speed of transportation in mode m (km/hour)* |
| $d_l^{np}$ | *The demand of wholesaler l for type-p processed product n (ton)* |
| $Cf_j$ | *Fixed cost of opening one processing center chamber (USD)* |
| $Cf_k$ | *Fixed cost of opening one storage center chamber (USD)* |
| $Costtr_m$ | *Cost of transportation per unit load per unit distance (USD.ton$^{-1}$.km$^{-1}$)* |
| $Corder_m$ | *Cost of ordering mode m of transportation (USD/ unit vehicle)* |
| $Cpro_n^p$ | *Cost of type-p processing of each unit of product n (USD.ton$^{-1}$)* |
| $Csto$ | *Cost of each unit of product to be stored (USD.ton$^{-1}$)* |

**Variables.**

| *Integer variables* | |
|---|---|
| $q_i^n$ | *Amount of product n purchased from farm i and entered the supply chain (ton)* |
| $q_{i \to m \to j}^n$ | *Amount of product n which is transported from farm i to processing center j by mode m (ton)* |
| $Extq_{in}$ | *Amount of product n purchased from farm i to be sold to the second market (ton)* |
| $q_{jnp}$ | *Amount of type-p processed product n produced at center j (ton)* |
| $q_{j \to m \to k}^{np}$ | *Amount of type-p processed product n transported from processing center j to storage center k by mode m (ton)* |
| $q_k^{np}$ | *Amount of type-p processed product n stored in storage center k (ton)* |
| $q_{k \to m \to l}^{np}$ | *Amount of type-p processed product n transported from storage center k to customer center l by mode m (ton)* |
| $q_l^{np}$ | *Amount of type-p processed product n delivered to customer l (ton)* |
| $t$ | *Time duration (hour)* |
| $N_{i \to m \to j}$ | *No. of mode-m vehicles for transferring harvested products from farm i to center j* |
| $N_{j \to m \to k}$ | *No. of vehicles of mode-m for transferring products from center j to center k* |
| $N_{k \to m \to l}$ | *No. of vehicles of mode-m for transferring products from center k to center l* |
| $ex_{jnp}$ | *Weight of processed product n in type-p over the demand of customer l (ton)* |
| $sh_{jnp}$ | *Weight of processed product n in type-p less than the demand of customer l (ton)* |
| *Binary variables* | |
| $X_{jnp}$ | *1, if the equipment for processing type-p product n is installed in center j; 0 otherwise* |
| $X_{knp}$ | *1, if the equipment for storing type-p product n is installed in center k; 0 otherwise* |

Profit objective function and constraints are described as follows:

**Profit objective function.** The objective function is defined to maximize the supply chain profit. It is equal to the revenue from both, selling products to customers and the second market minus the total supply chain and quality costs (Eq 2).

$$\textbf{Profit} = \textbf{Total revenue} - \textbf{Total cost}$$

$$
\begin{aligned}
= &\left[ \sum_1^l \sum_1^n \sum_1^p sa_n^p \cdot (q_l^{np} - ex_l^{np}) + \sum_1^i \sum_1^n sa'_n \cdot (pd_i^m - q_{ni}) \right] \\
&- \left[ \sum_1^i \sum_1^n pri_n \cdot q_{ni} + \sum_1^i \sum_1^l \sum_1^n \sum_1^p pri_n \cdot (d_l^{np} - q_{ni}) + \sum_1^i \sum_1^l \sum_1^n \sum_1^p pri'_n \cdot (pd_i^n - d_l^{np}) \right] - \sum_1^j \sum_1^n \sum_1^p Cf_j \cdot X_{j^{np}} \\
&+ \sum_1^k \sum_1^n \sum_1^p Cf_k \cdot X_{k^{np}} \right] - \sum_1^j \sum_1^n \sum_1^p Cpro_n^p \cdot q_{j^{np}} \right] \\
&- \left[ \sum_1^k \sum_1^n \sum_1^p Csto \cdot q_{k^{np}} \right] - \left[ \sum_1^m Costtr_m \cdot (\sum_1^i \sum_1^n \sum_1^j q_{ni \rightarrow mj} \cdot d_{ij} + \sum_1^j \sum_1^n \sum_1^p \sum_1^k q_{j \rightarrow mk}^{np} \cdot d_{jk} \right. \\
&+ \sum_1^k \sum_1^n \sum_1^p \sum_1^l q_{k \rightarrow ml}^{np} \cdot d_{kl}) + \sum_1^m Corder_m \cdot (\sum_1^i \sum_1^j N_{i \rightarrow mj} + \sum_1^j \sum_1^k N_{j \rightarrow mk} \\
&+ \sum_1^k \sum_1^l N_{k \rightarrow ml})] - \left[ \sum_1^i \sum_1^j \sum_1^m \sum_1^n sa_n^{(p=1)} \cdot q_{i \rightarrow mj}^n \cdot k_n^p \cdot \frac{d_{ij}}{v_m} \sum_1^j \sum_1^k \sum_1^m \sum_1^n \sum_1^p sa_n^p \cdot q_{j \rightarrow mk}^{np} \cdot k_n^p \cdot \frac{d_{jk}}{v_m} + \sum_1^k \sum_1^l \sum_1^m \sum_1^n \sum_1^p sa_n^p \cdot q_{kml}^{np} \cdot k_n^p \cdot \frac{d_{kl}}{v_m}) \right] \\
&- \left[ \sum_1^l \sum_1^n \sum_1^p sa_n^p \cdot sh_{l^{np}} \right] - \left[ \sum_1^l \sum_1^n \cdot \sum_1^p sa_n^p \cdot ext_{l^{np}} \right]
\end{aligned}
$$

(2)

Revenue consists of: 1) that obtained by selling the supplied demanded product, which is equal to the unit price of the sold product multiplied by the customer met demand; the latter equals the amount supplied in the supply chain minus that over the customer demand, and 2) that obtained by selling: a) the supply chain-decided products and b) in-excess products sent to the second market which is equal to the price of each unit of the low-quality product multiplied by the amounts in a and b.

Costs relate to: 1) purchasing high-quality (on contract) and low-quality (in-excess) products from farmers (with their own related prices), 2) locating processing and storage centers, 3) processing operations, 4) storing products in storage centers, 5) different supply chain distances (ton-km), 6) ordering different transportation modes, 7) revenue lost due to reduced product quality, 8) credit lost due to unmet demand and 9) unsold wasted product.

## Constraints

**Quantities equations.**

$$q_{ni} \leq pd_n^i \qquad ; \forall i, n \tag{3}$$

$$q_{ni} = \sum_1^j \sum_1^m q_{i \rightarrow mj}^n \qquad ; \forall n, i \tag{4}$$

$$\sum_1^i \sum_1^m \sum_1^n q_{i \rightarrow mj}^n = \sum_1^n \sum_1^p q_{j^{np}} \qquad ; \forall j \tag{5}$$

$$\sum_1^i \sum_1^m \sum_1^j q_{i \rightarrow mj}^n = \sum_1^j \sum_1^m \sum_1^k \sum_1^p q_{j \rightarrow mk}^{np} \qquad ; \forall n \tag{6}$$

$$q_{j^{np}} = \sum_1^k \sum_1^m q_{j \rightarrow mk}^{np} \qquad ; \forall j, n, p \tag{7}$$

$$\sum_1^i \sum_1^m q_{i \rightarrow mj}^n = \sum_1^k \sum_1^m \sum_1^p q_{j \rightarrow mk}^{np} \qquad ; \forall j, n \tag{8}$$

$$\sum_1^j \sum_1^m q_{jmk}^{np} = \sum_1^l \sum_1^m q_{kml}^{np} \qquad ; \forall k, n, p \tag{9}$$

$$\sum_{1}^{k}\sum_{1}^{m}q_{k\to ml}^{np}= q_{l^{np}} \qquad\qquad ;\forall \mathbf{l}, \mathbf{n}, \mathbf{p} \qquad\qquad (10)$$

$$extq_{i^{n}} = pd_{n^{i}} - q_{n^{i}} \qquad\qquad ;\forall \mathbf{n}, \mathbf{i} \qquad\qquad (11)$$

Constraints (3) to (10) ensure the product weight in different supply chain steps—from the farm to the customer (considering the amount of the farm production). Constraint (11) addresses in-excess low-grade products to be sold in the second market; these are produced, but not delivered to customers through the supply chain for different reasons.

$$q_{j}^{np} \leq cap_{j^{p}}.X_{j^{np}} \qquad\qquad ;\forall \mathbf{j}, \mathbf{n}, \mathbf{p} \qquad\qquad (12)$$

$$q_{k}^{np} \leq cap_{k^{p}}.X_{k^{np}} \qquad\qquad ;\forall \mathbf{k}, \mathbf{n}, \mathbf{p} \qquad\qquad (13)$$

$$X_{j}^{np} \leq q_{j^{np}} \qquad\qquad ;\forall \mathbf{j}, \mathbf{n}, \mathbf{p} \qquad\qquad (14)$$

$$X_{k}^{np} \leq q_{k^{np}} \qquad\qquad ;\forall \mathbf{k}, \mathbf{n}, \mathbf{p} \qquad\qquad (15)$$

Constraints (12) to (15) indicate that quantities of processed and stored products, respectively, in activated processing and storage centers are determined based on the capacities of these centers. If centers are not active, the quantities would be zero.

**Travel time in the supply chain equation.**

$$t = {^{d}}/_{v} \qquad\qquad (16)$$

Constraint (16) indicates that the vehicles used in the transportation system of the supply chain have uniform speeds.

**Number of vehicles.**

$$N_{i\to mj} = \sum_{1}^{n}(q_{i\to mj}^{n}/\boldsymbol{cap}_{\boldsymbol{m^{(p=1)}}}) \qquad ;\forall \mathbf{i}, \mathbf{m}, \mathbf{j} \qquad\qquad (17)$$

$$N_{j\to mk} = \sum_{1}^{n}\sum_{1}^{p}(q_{j\to mk}^{np}/\boldsymbol{cap}_{\boldsymbol{m^{p}}}) \qquad ;\forall \mathbf{j}, \mathbf{m}, \mathbf{k} \qquad\qquad (18)$$

$$N_{k\to ml} = \sum_{1}^{n}\sum_{1}^{p}(q_{k\to ml}^{np}/\boldsymbol{cap}_{\boldsymbol{m^{p}}}) \qquad ;\forall \mathbf{k}, \mathbf{m}, \mathbf{l} \qquad\qquad (19)$$

$$N_{m} = \sum_{1}^{i}\sum_{1}^{j}N_{i\to mj} + \sum_{1}^{j}\sum_{1}^{k}N_{j\to mk} + \sum_{1}^{k}\sum_{1}^{l}N_{k\to ml} \qquad\qquad ;\forall \mathbf{I}, \mathbf{j}, \mathbf{k}, \mathbf{m}, \mathbf{l} \quad (20)$$

Constraint (17) to (20) determines needed vehicles in different modes to transfer materials in different supply chain steps and the whole supply chain assuming full-capacity active vehicles.

**Shortage and extra quantities constraints.**

$$d_{l^{np}} - q_{l^{np}} = sh_{l^{np}} - ext_{l^{np}} \qquad\qquad ;\forall \mathbf{l}, \mathbf{n}, \mathbf{p} \qquad\qquad (21)$$

$$ext_{l^{np}} \leq \mathbf{M}.\mathbf{y} \qquad\qquad ;\forall \mathbf{l}, \mathbf{n}, \mathbf{p} \qquad\qquad (22)$$

$$sh_{l^{np}} \leq \mathbf{M}.(\mathbf{1-y}) \qquad\qquad ;\forall \mathbf{l}, \mathbf{n}, \mathbf{p} \qquad\qquad (23)$$

Constraints (21) to (23) determine the in-excess and shortage amounts.

$$q_{n^i}, q_{i \to mj}^n, q_{j^n}, q_{j^{np}} q_{j \to mk}^{np}, q_k^{np}, q_{k \to ml}^{np}, q_l^{np}, Ext_l^{np}, Sh_l^{np}, Extq_{i^n}, N_{i \to mj}, N_{j \to mk}, N_{k \to ml}, N_m, \geq 0 \; and \; Integers \quad (24)$$

$$X_{j^{np}}, X_{k^{np}} \in \{0, 1\} \quad (25)$$

Constraints (24) and (25) illustrate non-negativity and binary variables.

## 4. Case study

In this section, we implement the proposed model in an Iranian raw and processed vegetable products' company, the Razian Company, as a case study. Iran country has been bestowed with a wide range of climate and physio-geographical conditions and as such is most suitable for growing various kinds of vegetables, its production of vegetables is increasing. Moreover, agricultural products are profitable fields for investment. Since Iran possesses a large variety of flora with manufacturers, in equal measure, analysis of the working of the vegetable market is critical [55]. There is an apparent shortage of related supply chain in Iran country. The goal of the case study is to evaluate the efficacy of the proposed model under real-world conditions and to address the needs of the firm in question. The case study used a four-echelon SCND, and materials were supplied, processed, and stored (echelons 1–3) in the firm area (origin) while the last-level centers were located all over the country; in addition, a center was established as a second market to collect the in-excess products, as shown in Fig 1. The mentioned lateral market imposes no costs on the supply chain because it is closest to farms, and customers pay the transportation costs.

At first, the firm seasonally provided the vegetables from the suppliers. Suppliers were specified and contracted in advance in fertilized source centers (i = 4) of selected vegetables (n = 3). The farm centers were, in *Kaboudrahang*, *Razan*, *Nahavand*, and *Malayer*, and the vegetable products were *Yarrow*, *Borage flower*, and *Melisa*. Secondly, the firm used the related processing on vegetables, or the products remained raw. There are potential processing center (j = 5) candidates in the case study. Thirdly, the firm stored the products in the storage centers for packaging. There are potential storage center (k = 5) candidates in the case study. The five potential processing and storage center candidates were *Kaboudrahang*, *Razan*, *Nahavand*, *Malayer*, and *Asadabad*. Finally, the firm delivered the demanded products to the customer centers. The customers were trade representatives of each province all over the country (l = 30). Due to the importance of the case study data for the application of the presented model, some were obtained from the enterprise resource planning (ERP) of Razian company [56]. In addition, data on fixed and variable costs of different transportation modes were obtained from the recent case study research done in Iran [39]. Data on the price of different raw and processed vegetable products were gathered from the statistics of the Ministry of Agriculture [57]. Details of the most critical data of the case study are presented in the table in S1 Table in the supporting information.

The designed mathematical mixed integer linear programming (MILP) model was implemented and solved using GAMS 24.1.2 software and an Intel 2.13-GHz processor by exact solution method by epsilon-constraint. The designed network, product type, amount (tons) produced and sent to, e.g., Tehran (Capital), the transportation mode at different supply chain levels, and amount (tons) delivered to the second market are shown in Fig 2.

### 4.1. Results

As shown in Fig 2, in the optimum point of maximizing profit by considering quality costs in perishable vegetables supply chain in the proposed MILP model, the processing, storage, and

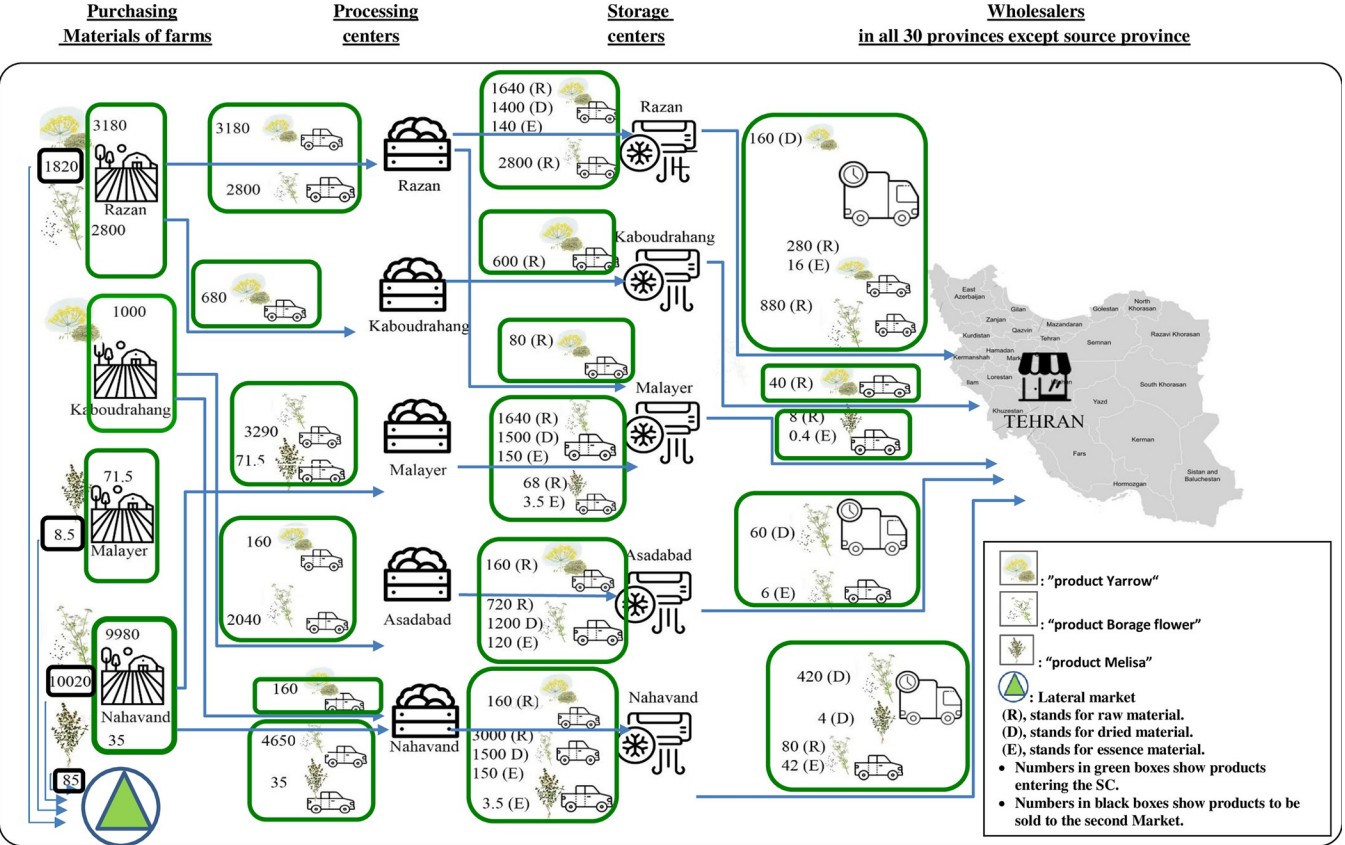

**Fig 2. Supply chain network design of multi-product (multi-vegetable) in Hamadan province (Quantities are in tons).** (R indicates Yarrow product, C indicates Borage flower product, T indicates Melisa product). Optimally, 564 trailers and 33068 trucks were needed in the designed supply chain network. Generally, in presented agricultural products' supply chain, some products have high quality-loss rates as well as demands for distances far from the cultivation center. This leads to a long post-harvest time for the product to reach the customer and, hence, a high rate of quality loss and a drop in the product price. This fact makes the supply chain decision maker set the lateral market due to not delivering those products to those customers and hence delivering them to the second market. It is considered newly in the present research due to make quality loss of products in the supply chain, the less, hence the profit the more.

distribution centers are settled in similar locations, spatially. It leads to set process-storage-transfer type of hub centers, in compliance with the supply chain network proposed by Khazaeli et al. [39]. Out of 5 potential processing and storage/transfer centers, the model found all for the supply chain No. of facilities based on the center capacity and its setup costs (related parameters are listed in the table in S1 Table). It is similar to the model proposed by De Keizer et al., in which to decrease transport time and hence decay in related supply chain design, the centers were decentralized, [9]. Therefore, more hubs were opened.

The model determined the amount and type of the delivered products between all supply chain levels, by the supply chain programming, and provided the information on the product (ton) if it was possible to supply to meet the customer demand. The details of provided products are presented in the table in S2 Table in the supporting file. Here, the supply chain management decides not to offer part of products to the customer and sells them at a second-grade price to the second market to maximize the chain profit by minimizing the quality loss-related cost along the chain (highlighted as unmet demands in the table in S2 Table). In such a case, saving the low-quality cost of the perishable product will bring more revenue for the chain.

The model also selected the center-to-center transportation mode considering the vehicle speed to reduce time and, hence, the quality degradation and transportation costs. The table in

S2 Table in the supporting file lists the number of each vehicle type required to transfer products. In result, the supply chain used trucks about 60 times more than trailers because of being faster. It used trailers, although with higher order costs, only in long distances, e.g., from storage centers to customer centers due to their more than ten times more capacity than trucks which led to fewer vehicle orders and, hence, less vehicle order costs. As shown in Fig 2, in all supply chain steps, except the last, the model suggests using trucks because of their higher speed than trailers and their less order costs than trailers (The vehicle-related parameters are shown in the table in S1 Table).

In this chain, some produced, but supply chain-decided undelivered to the supply chain were sold to the second market with price of high-grade products. The products produced more than that guaranteed in the farmer's purchase contract, were sold to the second market with a much cheaper price (0.3 that of high-grade products). Both, amounted to 1820 tons of product *Yarrow* in *Razan*, 10020 tons of product Borage flower in *Nahavand* and 93.5 tons of product *Melisa* in *Malayer* and *Nahavand*, all were delivered to the second market.

Demands for all types of products were met except for fresh products, for which the demands were responded in centers closer to the previous echelon due, maybe, to their higher corruptibility and quality-loss rate than other types of products (The table in S1 Table in the supporting information lists the perishability rate of each processed products than the fresh one) and, hence, a price decline that makes them uneconomical to deliver to customers.

## 4.2. Benefit and quality loss of the products in the supply chain

In the designed supply chain, as shown for the optimum solution point in Fig 3A, the revenue and total cost are, respectively, 27.3 and 18.5 million USD; therefore, the benefit is 8.8 million USD. The final product quality and quality loss in the supply chain are 28,357 and 643 (Unit of quality), respectively (Fig 3B).

The revenue of the supply chain (27.3 million USD) is due to: 1) selling the chain-demanded supplied products 21.9 (Million USD), 2) selling products not supplied to the chain and sold to the secondary market based on the chain management decision 0.08 (Million USD) and 3) selling products supplied more than that specified in the contract 5.32 to the secondary market (Million USD) (Fig 4A).

The revenue of farmers as main stakeholders, is 12.5 million USD, which goes to them by selling: 1) contract-demanded products delivered to the supply chain (10.2 million USD), 2) contract-demanded products supply chain-decided undelivered products (2.07 million USD) and 3) in-excess-of-contract products to the second market (0.24 million USD) (Fig 4B).

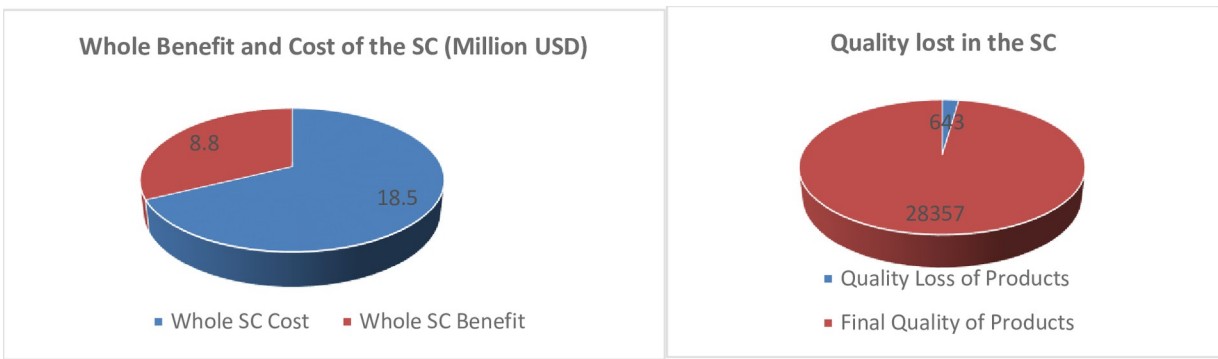

**Fig 3.** (a). Profit/ cost of the SC designed. (b). Final quality/ quality losses in the SC design.

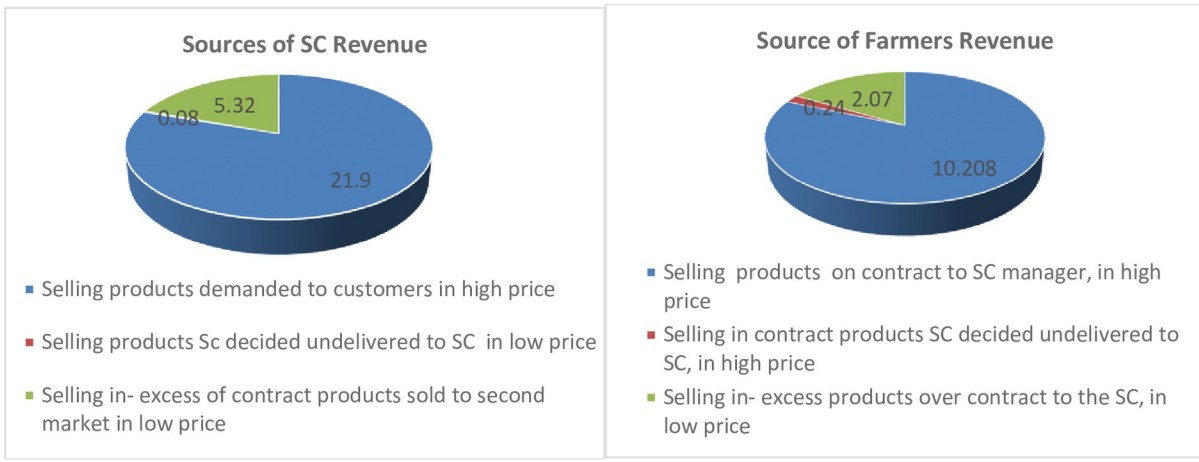

**Fig 4.** (a). Supply chain revenue parts. (b). Farmers' revenue parts.

Total supply chain benefit (8.8 million USD) comes from supplying products to customers considering the demand (5.7 million USD) and products to the second market (3.1 million USD). In addition, the total revenue of farmers is (12.5 million USD) (Fig 5).

## 4.3. Supply chain cost breakdown considering quality cost and other supply chain costs

The supply chain cost (18.5 million USD) consists of 8 elements, among which purchasing, including buying raw materials for the supply chain (10.2 million USD) and in-excess materials (2.3 million USD) for selling to the second market, is the costliest, and revenue lost due to reduced product quality along the chain (5.4 million USD) stand next. Other costs in the case studied, in the order of higher values, include quality cost of unmet demand of fresh products in long distances (0.38 million USD), processing (0.1 million USD), logistic transportation (0.07 million USD), storage (0.03 million USD), establishing facility centers (0.02 million USD); product waste has zero cost. The percent share of total costs, including those of the network, supply chain logistics and quality costs is shown in Fig 6.

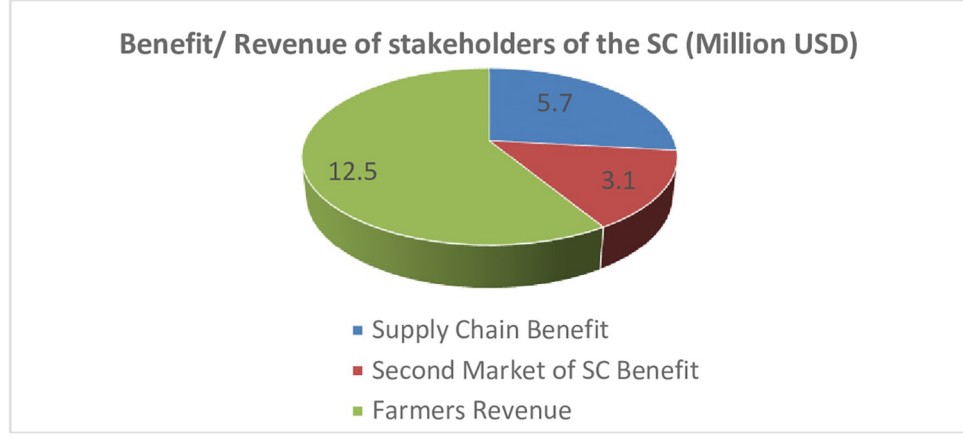

**Fig 5. Revenue of farmers and network designed profits.**

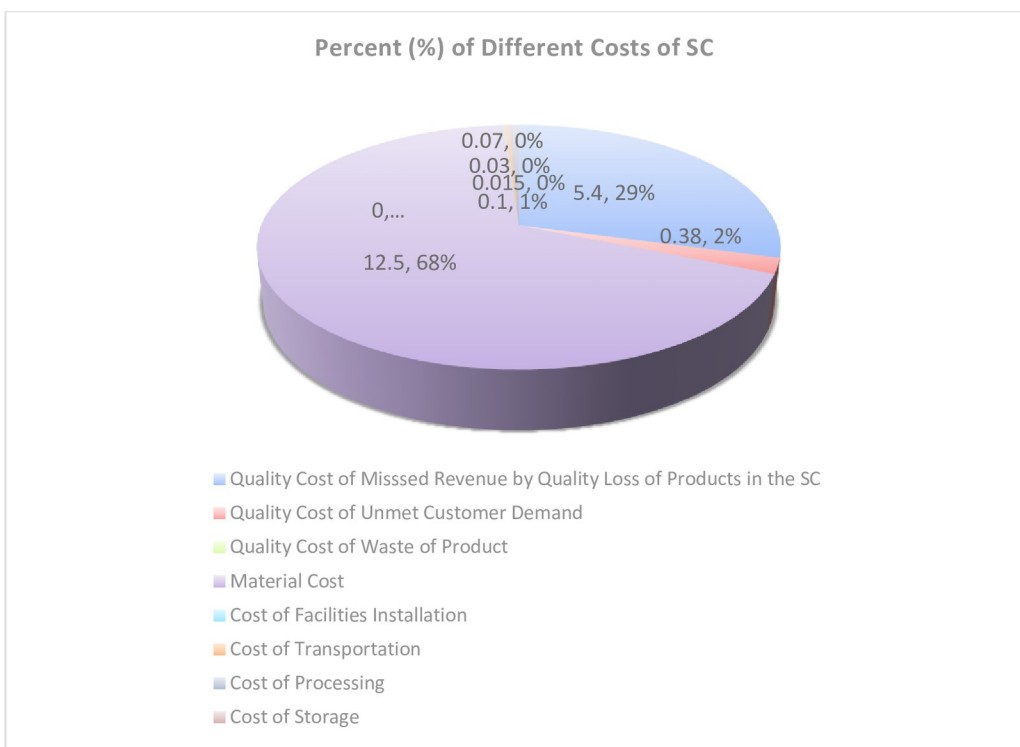

**Fig 6. Supply chain different costs.**

As shown in Fig 6, 29% of the costs (5.4 million USD) in the supply chain of perishable product supply relate to the revenue lost due to the product quality loss by unmet fresh products. On the other hand, the quality cost of unmet demand for fresh products in long distances is 0.38 million USD. The most part of the mentioned costs are compensated by revenue earned by selling these products to the second market by 5.32 million USD.

The designed supply chain has other profits, which are: 1) preventing low-quality products from being produced at the request of the chain customers and 2) sending products produced over that specified in the contract (due to unpredicted agricultural products produced) to the secondary market and, hence, preventing them from entering the environment as waste.

The model accuracy was verified by changing its parameters and examining its responses to the changes. The validity of the proposed model has also been confirmed by comparing the results of the present SCND, with a vicinity secondary market (Fig 2), and those of the existing chain, without such a market. Related experts have evaluated the proposed model, validated it, and concluded that the chain profit has increased due to its reduced quality costs. The sensitivity analysis is presented to evaluate the effect of changing some parameters on variables and the objective function, in the following.

### 4.4. Sensitivity analysis

Parameters to which model responses investigated in reaction, are the reaction rate of products and speed of different transportation modes as they relate to the quality loss of products and cost of supply chain during the time after harvest. Model responses to changes have been analyzed and explained orderly in the following:

**Quantity of products and revenue versus reaction rate (k) of products.** The quality loss rate (k) of different products varies depending on their reactivity, and processing reduces this

rate in fresh products. To prevent the quality cost resulting from the products' quality loss and price decline, the chain provides just part of the fresh product demands, not far than a specific distance (The table in S2 Table in the supporting information). When the quality loss rate (k) changes, the amount of the customer-demanded met products as well as those not enter the chain change too; the latter are processed at the beginning of the chain immediately after they are purchased and then sold as low-grade products to the second market. The ratio of the customer-offered to customer demand for different types of products and the amount sold to the second market were examined considering the product quality loss rate (k). The effects of the quality loss rate (k) on the stakeholders' profit and revenue have also been studied. A summary of the results is shown in Fig 7.

In the current chain, 96% of the demand for fresh products is met, and the rest is sold to the second market. As shown in Fig (7a), an increase in the quality loss rate (k) reduces the amount of fresh products. It increases the amount of those sold to the second market and supplied before entering the chain due to a sharp drop in fresh products, undesirability for customers, quality loss and price drop in the chain over time. A more increase in the mentioned rate (twice more) reduces the meeting rate of the customer-demanded fresh product from 96% to 35%; products sold to the second market increase from 64% to 100%, and the processed, dried and essence products, fully met, remain unchanged. Moreover, as shown in Fig (7b), an increase in the rate of product quality loss (k) does not reduce the farmer revenue, because the contract-specified products are bought from farmers at the original price.

As shown in Fig (7b), an increase in the quality-loss rate (k) of perishable products reduces the chain profit because some of these products, purchased from the farmer at the original contract price, do not enter the chain and are sold in the secondary market at lower prices

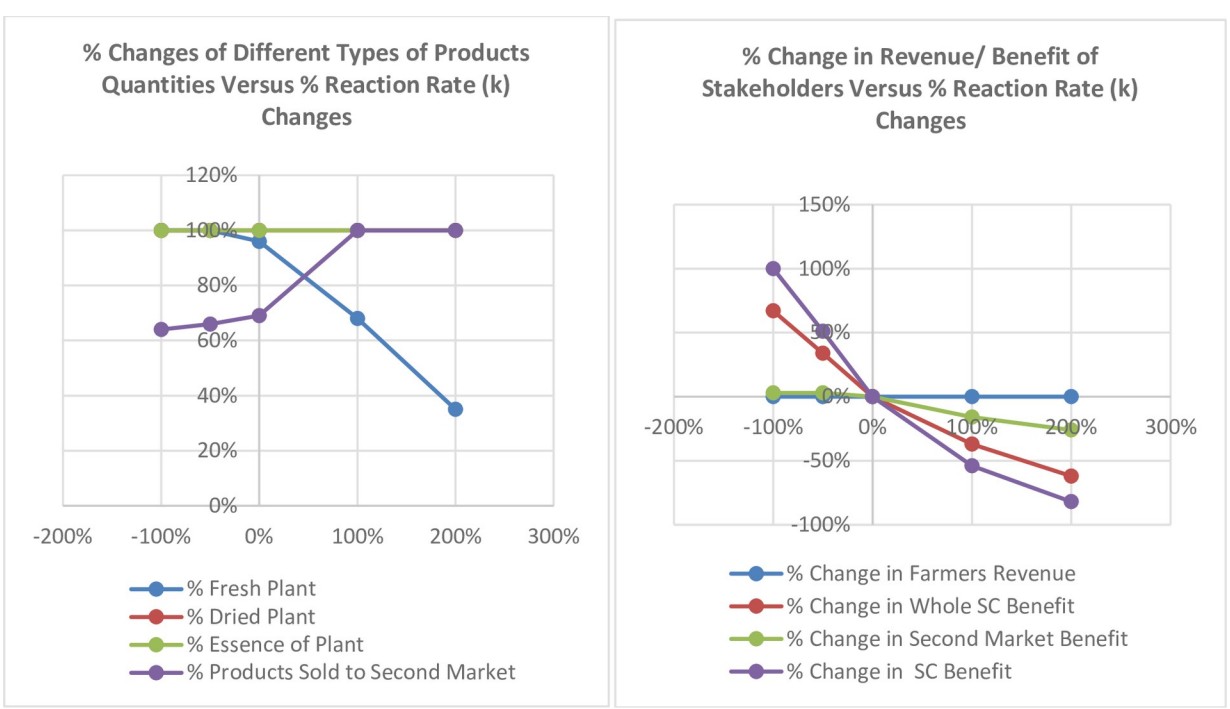

*(a) Changes in quantities of products. (b) Changes in revenue/ profit of farmers and SC parts.*

**Fig 7. Analysis of changes in reaction rate (k).** (a) Changes in quantities of products. (b) Changes in revenue/ profit of farmers and SC parts.

(here, 0.3 times the contract price). Therefore, considering higher quality-loss rates (k) in the SCND will result in sharper reduced profits for the supply chain and the secondary market.

**Supply chain cost/revenue versus speed of vehicles (v) changes.** Under present conditions and the speed (v) of the current fleet in the case study ($V_{trailer} = 80$ and $V_{truck} = 100$ (km/hour)), the model meets 96% of the demand for fresh products and all that for the dried and essence products; Faster fleet speeds enable more demands to be met (Fig 8A).

Increasing the speed (v) up to 50% will help the demand for fresh products to be met up to 100% and that for other products stays constant at 100%; however, reducing it up to 80% will not change the amount of processed products, but will cause the amount of the freshly supplied products to reach about 20% (Fig 8a).

As shown in Fig (8b), increasing the speed (v) leads to more use of faster vehicles (here, trucks). As shown, increasing the speed (v) to 100% will increase the number of needed trucks by 2%, but will not change the number of needed trailers. As mentioned earlier, trailers are used for outside-province long distances to respond to customers located far from the supply center. This will result in lower total long-distance transportation costs than trucks due to lower ton-km costs despite higher-order costs (The table in S1 Table in the supporting file).

Fig (8c) shows the minor increases in transportation costs and a noticeable reduction in the unmet-demand lost revenue due to the increased vehicle speed (v). Increasing the speed (v) up to 100% will increase the transportation costs by 11%, but reduces the unmet-demand lost revenue by 100%. This increased transportation cost of 0.008 million USD will prevent a revenue loss of 0.34 million USD, which is quite a significant figure.

It demonstrates that increasing the speed (v) will increase the number of vehicles, hence increase the transportation costs and the responded demand and ultimately prevent the revenue loss. Hence, increasing the speed (v) will lead to increased costs and enhanced chain revenue (Fig 9).

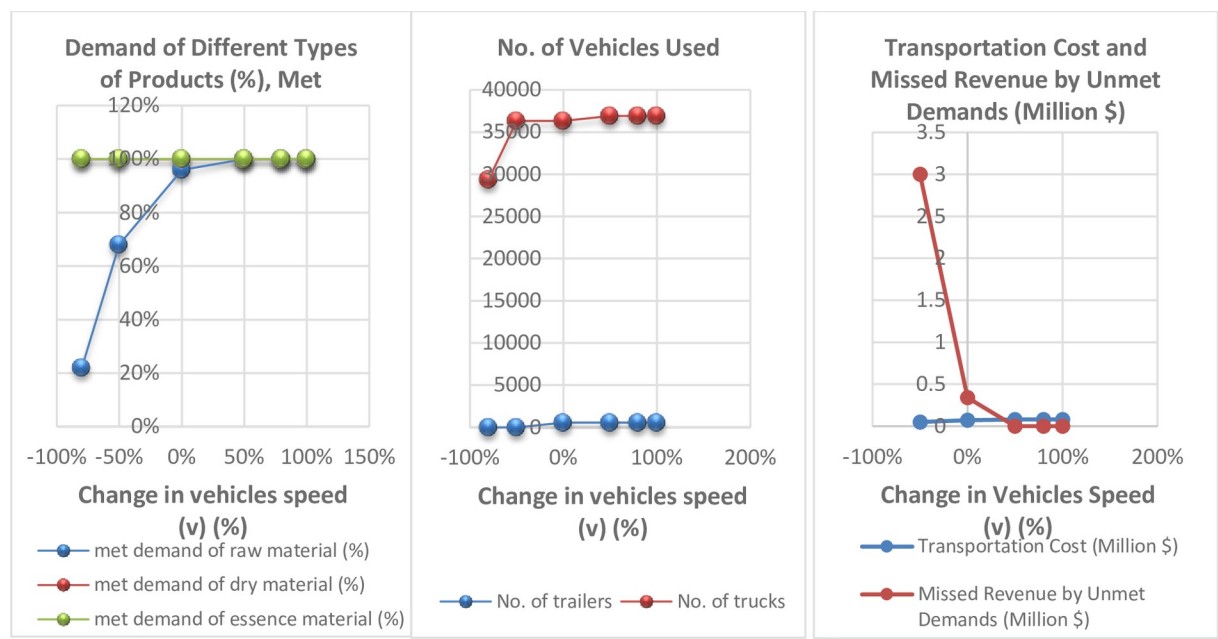

*(a) Change in quantities of products (b). Change in number of vehicles (c). Change in cost/ revenue*

**Fig 8. Analysis of supply chain designed results versus change in vehicles speed (v).** (a) Change in quantities of products (b). Change in number of vehicles (c). Change in cost/ revenue.

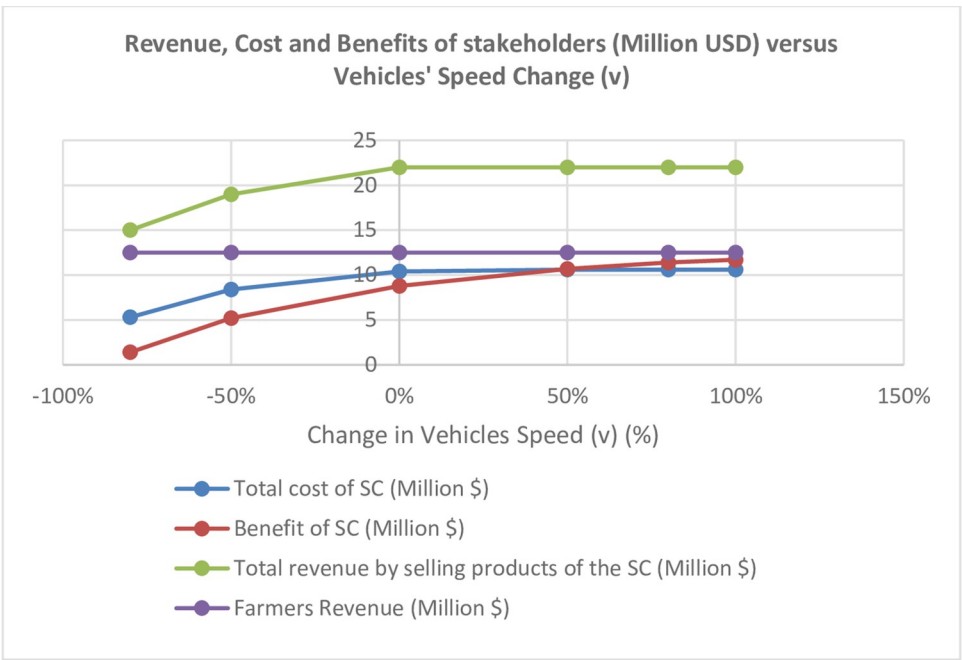

**Fig 9. Analysis of different stakeholders' cost/ revenue/ profit versus vehicles' speed (v) change.**

Since the increased revenue is greater than the increased cost, increasing the speed (v) will increase the chain profit; increasing the speed (v) up to 100% will increase the profit by 2.9 million USD (increased by 100%). Increasing the speed (v) will not affect the farmers' revenue. The results comply with the findings of Patidar and Agrawal in research on traditional agricultural chains in India, in which the transportation share in supply chain costs reached about 92% in the distribution sector [12]. It shows the importance of transportation strategies in this sector.

A comparison of designed supply chain with traditional supply chain in the case study is demonstrated in Fig 10.

Regarding demands for fresh products, as shown in Fig 10, their amounts in the two cases (with and without a secondary market) are 10868 and 10068 tons, respectively, showing an increase of about 0.08 times; this leads to a quality increase and, hence, customer satisfaction and profit increase. In both cases, demands for dry and essence products are fully satisfied. The comparison between the results of the present supply chain design in the case study and the results with the lateral market indicates that a lateral market in the supply chain will increase the chain profit and farmer income. However, in the optimal mode in this case study, they are increasing from 9.7 and 5.85 to 12.5 (about +50%) and 8.8 (about +20%), respectively.

The results show that newly designed supply chain is applicable in the field of the perishable products supply chain. It confirms the necessity of supplying innovative products of perishable ones such as processed agricultural products to meet new customer needs in a lateral market to the competitiveness. It complies with the findings of Malynka and Perevozova, who proposed the lateral markets in mature and immature markets in the brand creation process [25].

## 4.5. Managerial insight

Some lessons and insights for managers are as follows.

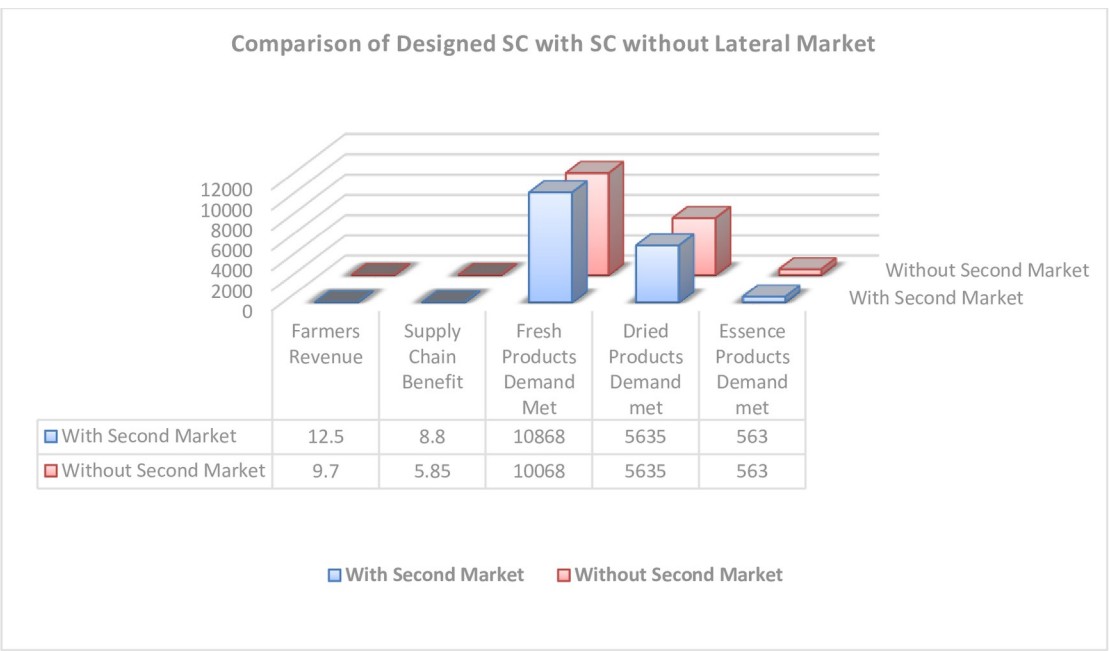

**Fig 10. Designed SC with a second market in comparison with SC without a second market.**

- All of the contract products are supposed to enter the chain; if not (for different reasons, e.g., chain management decision), some are sent to the second market, the presence of which prevents the produced products and resources (land, labor, energy, etc.) spent for those not entering the chain (for different reasons) from being wasted.

- Not considering a second market for fresh perishable agricultural products e. g., vegetables will lead to ignoring the post-harvest quality loss-related costs.

- Increasing the fleet speed is of great benefit to all the chain stakeholders including customers, chain management and environment because, on the one hand, it leads to increased response to the customer demand for more fresh products and selling customer-demanded high-quality products at prices proportionate to their grades will increase the chain profit and, hence, the total revenue, on the other, it prevents environmental pollution by letting more products to enter the supply chain and preventing wastes to be generated.

## 5. Conclusions

In this paper, a new approach is presented to optimize a logistics network design for distributing multiple products that are highly perishable and sensitive in quality and health of products to consumers, such as vegetables. Echelons of supply chain design include supply, processing, storage and customer. Considering the unpredictable amount of production of agricultural products and their perishability post-harvest, the second market which is accompanied by processing technologies to produce innovative products from the perishable products has been considered in the related supply chain network design, beside the main chain. The supply chain network design has been done based on maximization of profit by considering different quality costs in the supply chain. Quality costs include those due to: 1) quality-loss price-drop, 2) product waste and 3) losing credit with the customer for not meeting the desired demand. Since the chain integrity of these types of products is essential, the integrated one considered

in this study is managed by the chain management deployed in the product supply center. Programming has been done based on the maximization of profit by the MILP model considering the quality costs of products in the supply chain. To evaluate the modeling, a case study was used on three vegetables cultivated and harvested in a fertile area in Iran country in September 2023. The model was subsequently validated by multiple sensitivity analyses performed on some of the essential parameters that had a greater effect on the results.

In this supply chain design, as it is demonstrated in Fig 2, different decisions have been made at strategic, tactical and operational levels in order to maximize profit by considering the costs of quality in the supply chain. Decisions made are on the location of processing centers and storage centers, and product flow allocations in the designed supply chain. Moreover, the model decides on the operations of processing after harvest, such as drying and extracting, which leads to mitigated products' quality decay. In the next echelon after processing in the supply chain, there are storage centers in which products are stored to be distributed to the retailers. In the tactical level of decision making, the presented model decides on the allocation of farmers to processing centers and also processing centers to storage centers, moreover the allocation of storage centers to the retailers as the customers, also the number of products produced by farmers enters the supply chain and remains to be supplied to the second market and not deployed in the supply chain is determined. In the operational level of decision-making, the quantity of products and mode of transportation between different levels in the supply chain have been determined to meet the customers' needs.

Results of this research were compared with those of related recent studies [9, 12, 25, 39]. The comparisons demonstrated good conformity, especially, in compliance with recent research in lateral besides vertical markets [25]. It seems innovative second markets are required to meet other parts of demand. Settling the lateral market seems strategic, especially in perishable products. The lateral market regulates supply and demand and helps reduce the quality-loss-related costs of the chain and responds to another part of the market that has specific customers.

## 6. Managerial implications

The proposed model is generic and can help managers in food quality, customer service, and other related operations as a tool to assist in decision-making in the perishable agricultural products supply chain. Specially, the research done can have the following applications:

- The decisions stemming from the presented model are determined based on the products' degradation pattern to maximize its quality. The decisions include supplier selection, supply chain design, processing technology deployment, and vehicle deployment.

- A second market besides the chain and not higher than the vertical one in supply-based products such as vegetables, may result in a considerable increase in the chain profit without changing the resources, no reduction in the farmer income for unpredictable amounts of agricultural products production, and no wasted products preventing the environment from being polluted.

- The usage of lateral marketing is relevant, as it is the most effective way of competition in mature markets. However, when chains are designed for perishable products for optimum profit, the demand for some products with high quality-loss rates is not met due too long distances from distribution centers (if it is met, high-quality costs will be imposed on the chain). The related products are processed for secondary customers and delivered to them in the second market.

- Increased perishability rate of agricultural products reveals the effects and necessity of second markets next to the chain.

- Although, high-speed shipping fleets are expensive, using them will increase the chain profit because they reduce the post-harvest travel time and, hence, reduce the quality-loss-related costs of perishable products significantly. This way, the demands of more customers are met, customer credit costs will be prevented and the supply chain management and customers will both be benefitted. By applying the proposed model in the perishable agricultural products supply chain, the products are sold in the second market to meet the lateral part of the market.

As a result, different stakeholders such as farmers, customers, the environment, and the owner of the supply chain may benefit from the new supply chain network design.

## 7. Future research and limitations

Our framework is limited in some respects. With that said, this modeling limitations serve as a platform for extending it in future researches. One primary limitation of the presented model is that it does not consider the uncertainty in the amount of customers' demand. Therefore, the proposed model does not work for the problem in uncertain conditions. Also, the proposed model in this research has been solved by the exact-type solving method of mathematical programming, which is proper for solving the small size of problems such as the studied case. Considering the limitations above, using mathematical models by uncertainty considerations in the supply chain parameters and applying meta-heuristic methods to solve medium and large-sized problems are suggested in the future research. From the managerial perspective, the presented research works by the assumption of that upstream suppliers, freight transportation, processing centers, and storage facilities are integrated and it needs to build alignment between their organizations to deploy the solutions proposed by the output of the proposed framework. For these efforts to be successful, for future research, it is suggested to study how to cooperate all parties involved in the supply chain, and design the coordination infrastructure in the supply chain to yield the positive effects of proposed supply chain network design, in practice.

## Supporting information

**S1 Table. Parameters of case study network design.**
(DOCX)

**S2 Table. Quantity (tons) of each product delivered to customers and number of vehicles in logistics of designed SC.**
(DOCX)

## Acknowledgments

The authors are indebted to Mr. Ja'fary, the manager of "*Razian*" Co. (https://razian.co/), for his invaluable help to gather data in the case study. Also, the authors are grateful to the two anonymous referees for their valuable comments, which have led to significant improvements in this paper.

## Author Contributions

**Conceptualization:** Hadi Sahebi.

**Methodology:** Sareh Khazaeli.

**Supervision:** Ramazan Kalvandi.

**Writing – original draft:** Sareh Khazaeli.

**Writing – review & editing:** Sareh Khazaeli.

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
