## [Decision Letter · Decision Letter 0]

30 Oct 2023

PONE-D-23-27530A Multi-level Multi-Product Supply Chain Network Design of Vegetables Products Considering Quality Costs: A Case StudyPLOS ONE

Dear Dr. khazaeli,

Thank you for submitting your manuscript to PLOS ONE. After careful consideration, we feel that it has merit but does not fully meet PLOS ONE’s publication criteria as it currently stands. Therefore, we invite you to submit a revised version of the manuscript that addresses the points raised during the review process.

We look forward to receiving your revised manuscript.

Kind regards,

Md. Monirul Islam, PhD

Academic Editor

PLOS ONE

Journal Requirements:

4. We notice that your supplementary tables are included in the manuscript file. Please remove them and upload them with the file type 'Supporting Information'. Please ensure that each Supporting Information file has a legend listed in the manuscript after the references list.

**Additional Editor Comments:**

Reviewer -1:

It should be recommended to add exact discussion about considering linear behavior for a perishable supply chain.

Some sensitivity analysis are so trivial and should be omitted.

Literature review should be checked. some significant published work has not been considered.

Comparative analysis should be covered difference between present condition of case study and proposed approach.

Reviewer-2:

Although the problem studied in this article is ok but the exposition is incomplete and inaccurate. Most of the viewpoints are simply explained abut not fully explained, and there are many problems in the model setting. Simplifying the model may get better results. Based on these concerns, I have to recommend that this paper should be considered for major revision.

While the paper provides a clear outline of the paper's objectives and methodology, it lacks specific details on the theories and approaches employed. To enhance the clarity and comprehensibility, I suggest incorporating the following :

1. Provide a brief summary of the approach: Explain the key principles and advantages of the method included to provide readers with a better understanding of its relevance to the proposed models.

2. Elaborate on the multiple-objective programming tools: Highlight the specific tools and techniques utilized and how they contribute to addressing the presence of such a huge network problem. This would help the readers comprehend the novelty and significance of the approach. Research gaps can be improved by using https://doi.org/10.3390/math9172093 and https://doi.org/10.1007/s11276-019-02246-6

https://link.springer.com/article/10.1007/s10100-023-00870-4

https://doi.org/10.1007/s10100-023-00874-0

3. Expand on the sufficient conditions to estimate the model: Provide further insights/assumptions into the conditions established by the proposed models, explain how these conditions are derived, and clarify their role in the overall analysis for the case data. Separate section to be considered for Case considerations.

4. Enhance the explanation of the application in the food sector/ perishable products : Give specific information about how the suggested models were used in the food sector/ perishable industry (please cite https://doi.org/10.1080/00207543.2012.752587), such as the data sources, variables that were looked at, and any results or insights that were gained from the use of the current data sets.

5. The results of the paper are simply stated without good explanations. The discussion of the obtained results must be well improved, highlighting the insights of the research findings and support from earlier literature such as https://doi.org/10.1038/s41598-022-26449-8.

6. It would be better if providing a more complete process instead of just states how to calculate it.

This submission looks like the student’s assignment and not a research paper .

7. Limitation of the work to be included along with the future research directions.

Reviewers' comments:

Reviewer's Responses to Questions

**Comments to the Author**

1. Is the manuscript technically sound, and do the data support the conclusions?

Reviewer #1: Partly

Reviewer #2: Partly

2. Has the statistical analysis been performed appropriately and rigorously? 

Reviewer #1: N/A

Reviewer #2: Yes

3. Have the authors made all data underlying the findings in their manuscript fully available?

Reviewer #1: No

Reviewer #2: No

4. Is the manuscript presented in an intelligible fashion and written in standard English?

Reviewer #1: Yes

Reviewer #2: No

5. Review Comments to the Author

Reviewer #1: It should be recommended to add exact discussion about considering linear behavior for a perishable supply chain.

Some sensitivity analysis are so trivial and should be omitted.

Literature review should be checked. some significant published work has not been considered.

Comparative analysis should be covered difference between present condition of case study and proposed approach.

Reviewer #2: Although the problem studied in this article is ok but the exposition is incomplete and inaccurate. Most of the viewpoints are simply explained abut not fully explained, and there are many problems in the model setting. Simplifying the model may get better results. Based on these concerns, I have to recommend that this paper should be considered for major revision.

While the paper provides a clear outline of the paper's objectives and methodology, it lacks specific details on the theories and approaches employed. To enhance the clarity and comprehensibility, I suggest incorporating the following :

1. Provide a brief summary of the approach: Explain the key principles and advantages of the method included to provide readers with a better understanding of its relevance to the proposed models.

2. Elaborate on the multiple-objective programming tools: Highlight the specific tools and techniques utilized and how they contribute to addressing the presence of such a huge network problem. This would help the readers comprehend the novelty and significance of the approach. Research gaps can be improved by using https://doi.org/10.3390/math9172093 and https://doi.org/10.1007/s11276-019-02246-6

https://link.springer.com/article/10.1007/s10100-023-00870-4

https://doi.org/10.1007/s10100-023-00874-0

3. Expand on the sufficient conditions to estimate the model: Provide further insights/assumptions into the conditions established by the proposed models, explain how these conditions are derived, and clarify their role in the overall analysis for the case data. Separate section to be considered for Case considerations.

4. Enhance the explanation of the application in the food sector/ perishable products : Give specific information about how the suggested models were used in the food sector/ perishable industry (please cite https://doi.org/10.1080/00207543.2012.752587), such as the data sources, variables that were looked at, and any results or insights that were gained from the use of the current data sets.

5. The results of the paper are simply stated without good explanations. The discussion of the obtained results must be well improved, highlighting the insights of the research findings and support from earlier literature such as https://doi.org/10.1038/s41598-022-26449-8.

6. It would be better if providing a more complete process instead of just states how to calculate it.

This submission looks like the student’s assignment and not a research paper .

7. Limitation of the work to be included along with the future research directions.

6. PLOS authors have the option to publish the peer review history of their article (what does this mean?). If published, this will include your full peer review and any attached files.

Reviewer #1: No

Reviewer #2: **Yes: **Professor (Dr.) Sadia Samar Ali

---

## [Author Response · Author response to Decision Letter 0]

18 Dec 2023

Editor points:

Dear editor,

Thank you for the comments which led us to make the paper ready to be sent the journal in its standards. We have done the changes in the paper as follows.

 We add authors’ information in the manuscript.

 We do the requirements of the journal in the paper to be the paper proper for the journal.

 We add data in an appendix in a supplementary file and mentioned it in a manuscript.

 We make ready three other files as, manuscript, manuscript with track changes and response to the reviewers’ comments and attached them in the revision process.

Point-by-Point Response Letter

Reviewer 1:

Dear reviewer,

Thank you for your insightful comments and suggestions which led us to improve the paper. We have modified the manuscript thoroughly according to your valuable comments and helpful suggestions. Please find the revised version of the paper enclosed. The following are our responses to your comments. Please note that the referees’ comments are written in green and our responses in black. Also, the by-one-by response to the comments are as follows. 

 It should be recommended to add exact discussion about considering linear behavior for a perishable supply chain.

Thank you for your comment. The linear behavior for a perishable agricultural products, independent of environmental factors such as temperature variations has been re-explained in the “literature review” as follows:

In practice the decrease of a single quality attribute can be approximated by one of the four basic types of mechanism which are zero order reactions having linear kinetics, Michaelis Menten kinetics, first order reactions having exponential kinetics and autocatalytic reactions with logistic kinetics (HAYAKAWA and TIMBERS, 1977; Varoquaux and Wiley, 1994). Although linear kinetics are relatively rare, Michaelis Menten kinetics are observed more frequently. The Micaelis Menten kinetics reduces to a linear one in the initial region of decay, which is the most important in quality assessment. Therefore the variable of quality in vegetables in the initial region of decay can be considered as the function of time post-harvest linearly (Tijskens and Polderdijk, 1996). It is shown in Equation 1. 

Q(t)=Q_0-k .t (1)

Where, Q0 is the initial quality, t is time and k is a degradation rate. In a dynamic environment, as the quality function works by the exponential kinetics, the degradation rate of quality (k) depends on the type of chemical synthesis, the activation energy of material and the gas constant by Arrhenius relation (Chang, 1900).

 Some sensitivity analysis are so trivial and should be omitted.

Thank you for your comment. Two last part of the sensitivity analysis section has been improved not to be trivial as follows:

Since the increased revenue is greater than the increased cost, increasing the speed (v) will increase the chain profit; increasing the speed (v) up to 100% will increase the profit by 2.9 million USD (increased by 100%). Increasing the speed (v) will not affect the farmers’ revenue. The results comply with findings of Patidar and Agrawal in research of traditional agricultural chains in India, in which the transportation share in SC costs reached about 92% in the distribution sector (Patidar and Agrawal, 2020). It shows the importance of transportation strategies in this sector. 

Regarding demands for fresh products, as shown in Figure 2, their amounts in the two cases (with and without a secondary market) are 10868 and 10068 tons, respectively, showing an increase of about 0.08 times; this leads to a quality increase and, hence, customer satisfaction and profit increase. In both cases, demands for dry and essence products are fully satisfied. The results show that new designed supply chain is applicable in the field of perishable products supply chain. It confirms the necessity of supplying innovative products of perishable ones such as processed agricultural products to meet new customer needs in a lateral market in order to the competitiveness. It complies with the findings of Malynka and Perevozova, who proposed the lateral markets in mature and immature markets in brand creation process (Malynka and Perevozova, 2019).

 Literature review should be checked. Some significant published work has not been considered 

Thank you for your comment. The related papers have been studied, hence the “Introduction” and “literature review” has been updated by relevant papers. Also, the recent quantitative models have been studied, as all are shown orderly on pages 2-5, as follows:

 In the introduction section:

Quality of vegetables is one of important measures to its customers due to quality deterioration rate of products which relates to the health of consumers (Forman et al., 2012). Time decay and shortages are a common phenomenon in products with short life cycles, and financial volatility necessitates more accurate characterization of inventory costs based on time-adjusted value (Ali et al., 2013). Moreover the quality deterioration often happens in traditional supply chains which, for the most part, are poorly planned 

Recently some strategies studied in supply chain management of perishable products due to control the perishability of products which are inventory management (Ali et al., 2021), reverse logistic management (Ali et al., 2020), pricing (Mohammadi et al., 2023) and robust optimization (Goli et al., 2023).

Today, lateral marketing is the most effective way of competing on the mature/immature markets, where micro-segmentation and plenty of brands don’t leave any space for new opportunities (Malynka and Perevozova, 2019).

 In the literature review section:

The optimal operation strategy is acquired based on product quality (Guo et al., 2021).

Making decisions in an integrated way will reduce costs compared to individual decisions made at each level (Salehi et al., 2020; Song and Wu, 2022), Pasha et al. studied an integrated bi-objective quality-based production-distribution agri-food MILP SC model in which profitability is maximized by defining the quality as a function of such decisions as the location of hubs and transportation strategy throughout the SC. Moreover, in the greenery SCs, De Keizer et al. presented a model on which decisions made on the greenhouse location (strategic) based on the plant’s lifetime in that location (De Keizer et al., 2017). As changes in the temperature and enthalpy levels change the food quality (Soysal et al., 2012), Khazaeli et al. and Rong et. al determined the temperature of distribution centers and deliveries so as to meet the expectations of different customers as the operational decision making in a supply chain programming (Khazaeli et al., 2023; Rong et al., 2011).

In a dynamic environment, as the quality function works by the exponential kinetics, the degradation rate of quality (k) depends on the type of chemical synthesis, the activation energy of material and the gas constant by Arrhenius relation (Chang, 1981). In the most of network designs, cost and benefit and quality factors are the most important factors that are to be optimized. Mostly, agri-food should make a logical balance between two topics of price reduction and customer service improvement (Soysal, Bloemhof-Ruwaard et al. 2012). Usually, the economic criteria are considered based on resource utilization and customer responsiveness; the latter means meeting part of the customer demand in due delivery time.. De Keizer et al. and Khazaeli et al. indicated quality of agricultural products causes cost in supply chain’s network designs. They showed that there is a conflict between factors of cost and quality (De Keizer et al., 2015; Khazaeli et al., 2023).

 Comparative analysis should be covered difference between present condition of case study and proposed approach.

Thank you for your feedback. As the following comparison between present condition of case study and proposed approach were suggested by the reviewer to be included in the paper, we have done our best to clarify this issue by Figure 10 and its descriptions as follows:

The comparison between results of present supply chain design in the case study and the results with lateral market indicates that a lateral market in the supply chain will increase the chain profit and farmer income. However, in the optimal mode in this case study, they are increasing from 9.7 and 5.85 to 12.5 (about +50%) and 8.8 (about +20%), respectively. 

The results show that new designed supply chain is applicable in the field of perishable products supply chain. It confirms the necessity of supplying innovative products of perishable ones such as processed agricultural products to meet new customer needs in a lateral market in order to the competitiveness. It complies with the findings of Malynka and Perevozova, who proposed the lateral markets in mature and immature markets in brand creation process (Malynka and Perevozova, 2019).… 

 

Reviewer 2:

Dear reviewer,

Thank you for the time you put in to evaluate our manuscript. Your feedback has been invaluable to our work. Please find the revised version of the paper enclosed. The following are our responses to your comments. Please note that the referees’ comments are written in green and our responses in black. Also, the by-one-by response to the comments are as follows. 

Although the problem studied in this article is ok but the exposition is incomplete and inaccurate. Most of the viewpoints are simply explained about not fully explained, and there are many problems in the model setting. Simplifying the model may get better results. Based on these concerns, I have to recommend that this paper should be considered for major revision. While the paper provides a clear outline of the paper's objectives and methodology, it lacks specific details on the theories and approaches employed. To enhance the clarity and comprehensibility, I suggest incorporating the following

We appreciate your positive feedback on the topic. We have modified the manuscript according to the reviewer’s helpful comments and suggestions. We’ve done our best to more precisely clarify the confusing and poorly written component. In our opinion, now the manuscript is more suitable to the Journal.

 Provide a brief summary of the approach: Explain the key principles and advantages of the method included to provide readers with a better understanding of its relevance to the proposed models.

Thank you for your comment. The present study -past researches relationship is shown in the “the research gap and contributions”, as follows:

A review of quantitative SC researches on the perishability of agri-food by considering quality costs in Section 2, outlines the gaps in the literature as follows:

1. Despite the importance of cost of qualities in designing SCs due to the perishability of the products, the cost of quality concept has not been widely incorporated by researchers in the design of agricultural products’ SCs.

2. No research has paid attention to the lateral market to look at the quality problems from the side in which covering some target customers.

3. Few researchers have considered benefits of several stakeholders of the agricultural SCs simultaneously. The stakeholders in agricultural Products’ SC are consumers, farmers, environment and society.

Due to the importance and necessity of developing supply chain management (SCM) from a larger perspective to provide a win-win situation for each participant in the SC, in this paper, we aim to develop a novel mathematical model in order to design a SC network, based on quality function elements in the vegetables’ sector. 

The main contributions are as follows:

 Providing a network design model for an integrated multi-level SC of perishable products wherein profit is optimized by considering quality decay aspect of the products.

 Optimizing the profit of the SC of perishable products considering different quality costs for them due to unmet demand, product waste and reduced revenue of low-quality products.

 Introducing a strategy of selling perishable products to lateral markets before letting products enter the chain to prevent the production of low-quality products along it. 

 Enabling the purchase of the farmer’s total agricultural product above the contract ceiling due to unpredictable production to prevent waste production and its scattering in the environment.

 Introducing a strategy of producing semi-processed, low-quality products (from those that did not enter the chain) to meet part of the market demand for lower-quality lower-price products.

The key principles and advantages of the method included to provide readers with a better understanding of its relevance to the proposed models are mentioned as follows.

The proposed SCND is a multi-product, multi-echelon model with exact (certain) demand that makes decisions at strategic, tactical and operational levels. It has focused on “quality” by considering the quality deterioration in a Micaelis Menten kinetics in the initial region of decay reduced to a linear one, moreover, by defining costs of quality degradation in the quality-cost functions.

The developed model is a four-echelon SC of perishable agricultural products in which static decade rate of the products is considered. Moreover, a lateral market is considered in the SC that does not stand higher than the vertical marketing and complete the main market. In the end, the developed model is applied to a case study of a firm in agricultural products’ industry with four echelons of farm-processing-distribution-customer centers. The vegetables selected as candidate for the present SC network design are Yarrow, Borage flower and Melisa, due to their priority in agricultural studies and their application in various industries (Agriculture ministry of Iran, 2022).

 Elaborate on the multiple-objective programming tools: Highlight the specific tools and techniques utilized and how they contribute to addressing the presence of such a huge network problem. This would help the readers comprehend the novelty and significance of the approach. Research gaps can be improved by using https://doi.org/10.3390/math9172093

Although there are some researches done in order to minimize quality losses of perishable products by multi-objective problem solving approaches (Ali et al., 2021, 2020; Goli et al., 2023; Khazaeli et al., 2023; Pasha et al., 2021), the programming in present research is a single objective problem solving by profit objective function underlying quality loss costs. Hence the proposed model is not a multi-objective, but also a single objective one.

The mentioned reference is cited as a related research in other parts of the manuscript, “the literature review”.

 Expand on the sufficient conditions to estimate the model: Provide further insights/assumptions into the conditions established by the proposed models, explain how these conditions are derived, and clarify their role in the overall analysis for the case data. Separate section to be considered for Case considerations.

Firstly, the conditions and assumptions of the modeling are presented in first part of section 3 as follows.

The designed supply chain of the firm is shown in Figure 1. 

Figure 1. Flow diagram of the agricultural products’ SC.

Source(s): Authors’ work

First, products through related contracts and in-excess products are bought from farmers in the study area. In the second echelon of the proposed SC, some or all of the purchased products are processed at related centers resulting in different degrees of product quality. Third, the products in former echelon are stored in cool storage centers until being distributed and fourth, they are sold to wholesalers. Another part of the purchased products are transferred to the second market as lower quality products in different industries (tea bags, spices in food, etc.). Different road modes of transport are used between different echelons of the SC. 

The modeling makes decisions at different echelons of the SC. Decisions made are (1) selecting farms and the quantity of raw products to be purchased from each of them, (2) the amount of products sold to the second market, (3) the number of processing and storage facilities to be settled in the SC, (4)

---

## [Decision Letter · Decision Letter 1]

2 Jan 2024

PONE-D-23-27530R1A Multi-level Multi-Product Supply Chain Network Design of Vegetables Products Considering Quality Costs: A Case StudyPLOS ONE

Dear Dr. khazaeli,

Thank you for submitting your manuscript to PLOS ONE. After careful consideration, we feel that it has merit but does not fully meet PLOS ONE’s publication criteria as it currently stands. Therefore, we invite you to submit a revised version of the manuscript that addresses the points raised during the review process. **Reviewer-1**

Respond to first comment "It should be recommended to add exact discussion about considering linear behavior for a perishable supply chain" is not acceptable because used references are old.

**Reviewer-2**

1. Consider adding an Implications section that zeroes in on managerial and policy-maker implications, as this may enhance the practical relevance of the research.

2. A dedicated section for Future Research and Limitations can provide readers with a clearer understanding of potential avenues for further inquiry and the boundaries of the current study.

3. Refraining from using bullets, especially in Future research, limitations, and uniqueness of the study can help maintain a consistent and professional format throughout the manuscript.

4. Why Quality as a keyword is used in the title? Replace with some other word.

We look forward to receiving your revised manuscript.

Kind regards,

Md. Monirul Islam, PhD

Academic Editor

PLOS ONE

Additional Editor Comments:

Dear Author

Thank you so much for your effort. Please provide the necessary corrections as suggested by the expert review panel. Good Luck.

Reviewers' comments:

Reviewer's Responses to Questions

**Comments to the Author**

1. If the authors have adequately addressed your comments raised in a previous round of review and you feel that this manuscript is now acceptable for publication, you may indicate that here to bypass the “Comments to the Author” section, enter your conflict of interest statement in the “Confidential to Editor” section, and submit your "Accept" recommendation.

Reviewer #1: (No Response)

Reviewer #2: (No Response)

2. Is the manuscript technically sound, and do the data support the conclusions?

Reviewer #1: Partly

Reviewer #2: Partly

3. Has the statistical analysis been performed appropriately and rigorously? 

Reviewer #1: N/A

Reviewer #2: Yes

4. Have the authors made all data underlying the findings in their manuscript fully available?

Reviewer #1: Yes

Reviewer #2: No

5. Is the manuscript presented in an intelligible fashion and written in standard English?

Reviewer #1: Yes

Reviewer #2: Yes

6. Review Comments to the Author

Reviewer #1: Respond to first comment "It should be recommended to add exact discussion about considering linear behavior for a

perishable supply chain" is not acceptable because used references are old.

Reviewer #2: I truly appreciate the effort dedicated to revising the manuscript. While reviewing the document, I have a few constructive suggestions for further improvement:

1. Consider adding an Implications section that zeroes in on managerial and policy-maker implications, as this may enhance the practical relevance of the research.

2. A dedicated section for Future Research and Limitations can provide readers with a clearer understanding of potential avenues for further inquiry and the boundaries of the current study.

3. Refraining from using bullets, especially in Future research, limitations, and uniqueness of the study can help maintain a consistent and professional format throughout the manuscript.

4. Why Quality as a keyword is used in the title ? Replace with some other word.

I hope these recommendations prove helpful in refining the manuscript.

7. PLOS authors have the option to publish the peer review history of their article (what does this mean?). If published, this will include your full peer review and any attached files.

Reviewer #1: No

Reviewer #2: No

---

## [Author Response · Author response to Decision Letter 1]

30 Jan 2024

Point-by-Point Response Letter

Reviewer 1:

Dear reviewer,

Thank you for your insightful comment which led us to improve the paper. We have modified the manuscript thoroughly according to your valuable comment. Please find the revised version of the paper enclosed. In newly enclosed revised version of the paper, the new changes in this second revised version of the manuscript are highlighted in yellow color. Track changes show the changes in the first revision of the manuscript.

The following is our response to your comment. Please note that the referees’ comment is written in green and our responses in black. 

 Respond to first comment "It should be recommended to add exact discussion about considering linear behavior for a perishable supply chain" is not acceptable because used references are old.

Thank you for your comment. The linear behavior for perishable agricultural products, independent of environmental factors such as temperature variations has been re-explained in the “literature review” with some new references as follows:

The quality function of perishable agricultural products can be either complex or simple (Centobelli et al., 2021). It has been shown that, the decrease of a single quality attribute of agricultural products can be approximated by one of the four basic types of mechanism which are zero-order reactions having linear kinetics, Michaelis Menten kinetics, first-order reactions having exponential kinetics, and autocatalytic reactions with logistic kinetics (HAYAKAWA and TIMBERS, 1977; Varoquaux and Wiley, 1994). For the concept of keeping quality, the actual mechanism does not play a role. For this reason, it is convenient to assume zero-order reaction kinetics (Tijskens and Polderdijk, 1996), and mostly the Michaelis Menten kinetics reduces to a linear one in the initial region of decay, which is the most important in quality assessment (Hertog et al., 2014). Therefore, the variable of quality in vegetables in the initial region of decay can be considered as the function of time post-harvest linearly, which is shown in Equation 1. 

Q(t)=Q_0-k .t (1)

Where, Q0 is the initial quality, t is time and k is a degradation rate. Borghi D de Freitas et al. minimized the losses of fruits and vegetables that happen during its storage in a mathematical model, based on Equation 1 (Ferreira et al., 2005).

Reviewer 2:

Dear reviewer,

Thank you for the time you put in to evaluate our manuscript. Your feedback has been invaluable to our work. Please find the revised version of the paper enclosed. In newly enclosed revised version of the paper, the new changes in this second revised version of the manuscript are highlighted in yellow color. Track changes show the changes in the first revision of the manuscript.

The following are our responses to your comments. Please note that the referees’ comments are written in green and our responses in black. Also, the by-one-by response to the comments are as follows. 

 Consider adding an Implications section that zeroes in on managerial and policy-maker implications, as this may enhance the practical relevance of the research.

Thank you for your comment. To enhance the practical relevance of the research, the managerial implications section is added as Section 6 to the end of the manuscript, as follows:

The proposed model is generic and can help logistic managers as a tool to assist in decision-making related to the logistics of perishable agricultural products in the supply chain. Specially, the research done can have the following applications:

 A second market besides the chain and not higher than the vertical one in supply-based products such as vegetables, may result in a considerable increase in the chain profit without changing the resources, no reduction in the farmer income for unpredictable amounts of agricultural products production, and no wasted products preventing the environment from being polluted.

 The usage of lateral marketing is relevant, as it is the most effective way of competition in mature markets.

 When chains are designed for perishable products for optimum profit, the demand for some products with high quality-loss rates is not met due too long distances from distribution centers (if it is met, high-quality costs will be imposed on the chain). 

 Increased perishability rate of agricultural products reveals the effects and necessity of second markets next to the chain.

 Although, high speed shipping fleets are expensive, using them will increase the chain profit because they reduce the post-harvest travel time and, hence, reduce the quality-loss-related costs of perishable products significantly. This way, the demands of more customers are met, customer credit costs will be prevented and the supply chain management and customers will both be benefitted. By applying the proposed model in the perishable agricultural products supply chain, the products are sold in the second market to meet the lateral part of the market. As a result, different stakeholders such as farmers, customers, the environment, and the owner of the supply chain may benefit from the new supply chain network design

 A dedicated section for Future Research and Limitations can provide readers with a clearer understanding of potential avenues for further inquiry and the boundaries of the current study.

A separate section has been dedicated to future research and limitations to provide readers with a clearer understanding of potential avenues for future research and the boundaries of the current study in Section 7 as follows:

The limitations of this study include not considering pricing policies. The proposed solution method computed the optimal solution by the exact-type solving method of mathematical programming with certain parameters and it does not work for the problem in uncertain conditions. Due to the size of the case study problem and related certain parameters, this solution method is not applicable to large-size problems. Considering the limitations of and the results obtained in this study, directions for future research include considering pricing policies for products sold to the second market to maximize the profit of the stakeholders. As it is computed the optimal solution by the exact-type solving method of mathematical programming with certain parameters, using mathematical models to consider uncertainties in the SC parameters is suggested. Moreover, applying meta-heuristic methods to solve medium and large-sized problems is suggested in future research.

 Refraining from using bullets, especially in Future research, limitations, and uniqueness of the study can help maintain a consistent and professional format throughout the manuscript.

Thank you for your comment. We refrain from using bullets in future research and limitations.

 Why Quality as a keyword is used in the title? Replace with some other word.

Thank you for your comment. The word “Quality” in the title is not a separate word and it is part of the concept of “Cost of quality”, which is introduced by Feigenbaum and Crozby in the Total Quality Management philosophy. “Cost of quality” concept is a technique to measure the non-conformance with quality features. To the above-mentioned concept be clearer, the title is modified as follows.

“A Multi-level Multi-Product Supply Chain Network Design of Vegetables Products Considering Costs of Quality: A Case Study”.

---

## [Decision Letter · Decision Letter 2]

4 Mar 2024

PONE-D-23-27530R2A Multi-level Multi-Product Supply Chain Network Design of Vegetables Products Considering Costs of Quality: A Case StudyPLOS ONE

Dear Dr. khazaeli,

Thank you for submitting your manuscript to PLOS ONE. After careful consideration, we feel that it has merit but does not fully meet PLOS ONE’s publication criteria as it currently stands. Therefore, we invite you to submit a revised version of the manuscript that addresses the points raised during the review process. Comments:1. The abstract needs to have a smooth flow in terms of the discussions . Don't include pointwise discussions.

2. Many subsections were included, which prevented the readers from having a smooth reading. Kindly adjust and remove the subsection numbers.

3. Data availability references and tables/data-based in-depth information are not given .

4. The paper needs clearer insights into future trends and the ability of researchers to pursue them based on existing research. More detailed discussions on future research directions, trends, challenges, and opportunities are needed.

Be sure to:Indicate which changes you require for acceptance versus which changes you recommendAddress any conflicts between the reviews so that it's clear which advice the authors should followProvide specific feedback from your evaluation of the manuscriptPlease ensure that your decision is justified on PLOS ONE’s publication criteria and not, for example, on novelty or perceived impact.

We look forward to receiving your revised manuscript.

Kind regards,

Md. Monirul Islam, PhD

Academic Editor

PLOS ONE

Journal Requirements:

Additional Editor Comments :

1. The abstract needs to have a smooth flow in terms of the discussions . Don't include pointwise discussions.

2. Many subsections were included, which prevented the readers from having a smooth reading. Kindly adjust and remove the subsection numbers.

3. Data availability references and tables/data-based in-depth information are not given .

4. The paper needs clearer insights into future trends and the ability of researchers to pursue them based on existing research. More detailed discussions on future research directions, trends, challenges, and opportunities are needed.

Reviewers' comments:

Reviewer's Responses to Questions

**Comments to the Author**

1. If the authors have adequately addressed your comments raised in a previous round of review and you feel that this manuscript is now acceptable for publication, you may indicate that here to bypass the “Comments to the Author” section, enter your conflict of interest statement in the “Confidential to Editor” section, and submit your "Accept" recommendation.

Reviewer #1: (No Response)

Reviewer #2: (No Response)

Reviewer #3: All comments have been addressed

2. Is the manuscript technically sound, and do the data support the conclusions?

Reviewer #1: Partly

Reviewer #2: Partly

Reviewer #3: Yes

3. Has the statistical analysis been performed appropriately and rigorously? 

Reviewer #1: N/A

Reviewer #2: N/A

Reviewer #3: Yes

4. Have the authors made all data underlying the findings in their manuscript fully available?

Reviewer #1: Yes

Reviewer #2: No

Reviewer #3: Yes

5. Is the manuscript presented in an intelligible fashion and written in standard English?

Reviewer #1: Yes

Reviewer #2: Yes

Reviewer #3: Yes

6. Review Comments to the Author

Reviewer #1: Respond of authors about linear behavior for a perishable supply chain is not suitable nowadays thence originality of work is weak.

Reviewer #2: 1. The abstract needs to have a smooth flow in terms of the discussions . Don't include pointwise discussions.

2. Many subsections were included, which prevented the readers from having a smooth reading. Kindly adjust and remove the subsection numbers.

3. Data availability references and tables/data-based in-depth information are not given .

4. The paper needs clearer insights into future trends and the ability of researchers to pursue them based on existing research. More detailed discussions on future research directions, trends, challenges, and opportunities are needed.

Reviewer #3: No comment

Good work

-----------------

------------------

-----------------

7. PLOS authors have the option to publish the peer review history of their article (what does this mean?). If published, this will include your full peer review and any attached files.

Reviewer #1: No

Reviewer #2: No

Reviewer #3: **Yes: **Peiman Ghasemi

---

## [Author Response · Author response to Decision Letter 2]

17 Apr 2024

Point-by-Point Response Letter

Academic Editor:

Dear academic editor,

Thank you for your attention to our manuscript. As you emphasized on the second reviewer questions, the questions you mentioned are answered in pages 3-4.

 

Reviewer 1:

Dear reviewer,

Thank you for your insightful comment which led us to improve the paper. We have modified the manuscript thoroughly according to your valuable comment. Please find the revised version of the paper enclosed. In newly enclosed revised version of the paper, the new changes in this third revised version of the manuscript are highlighted in yellow color. Track changes show the changes in the second revision of the manuscript.

The following is our response to your comment. Please note that the referees’ comment is written in green and our responses in black. 

 Respond of authors about linear behavior for a perishable supply chain is not suitable nowadays thence originality of work is weak.

Thank you for your comment. The linear behavior for perishable agricultural products is widely used in the literature in modeling of the products’ perishability. It has been re-explained in the “literature review”, precisely. Moreover, some new references, which uses this modeling approach in the quality definition of perishable agricultural products are mentioned to demonstrate that the application of the linear behavior of the perishable products’ quality in time has been recently used in different mathematical modeling of quality behavior, as follows:

The quality function of perishable agricultural products can be either complex or simple [44]. It has been shown that, the decrease of a single quality attribute of agricultural products can be approximated by one of the four basic types of mechanism which are zero-order reactions having linear kinetics, Michaelis Menten kinetics, first-order reactions having exponential kinetics, and autocatalytic reactions with logistic kinetics [45], [46]. For the concept of keeping quality, it is convenient to assume zero-order reaction kinetics [47], and mostly the Michaelis Menten kinetics reduces to a linear one in the initial region of decay, which is the most important in quality assessment [48]. Therefore, the quality variable of vegetables in the initial region of decay can be considered in a widely used equation, in which the quality function changes by the time linearly. It is shown in Equation 1.

dQ/dt=k Q(t)=Q_0-k .t (1)

Where, Q0 is the initial quality, t is time and k is a degradation rate. In a dynamic environment, the well-known Arrhenius equation shows that the degradation rate (k) depends on the activation energy of the material, and the environmental factors [47], [49], [50]. 

The perishable products’ quality model shown in Equation 1 has been frequently used to capture the degradation of food products over time. For example, in the grocery retail chain, Wang and Li presented a pricing model to maximize food retailer’s profit in a dynamically identified food shelf life by using Equation 1 [51]. Chen and Chen proposed an on-site direct-sale dynamic supply chain inventory model, considering time-dependent quality losses for perishable foods [22]. Lejarza and Baldea presented a closed-loop, feedback-based control framework, that employs real-time product quality measurements for optimal supply chain management [52]. Moreover, Xu et al. presented a real time decision support framework to mitigate the quality degradation in the journey of agricultural perishable products from farm to the retailer in the supply chain based on the Equation 1 [53]. 

Reviewer 2:

Dear reviewer,

Thank you for the time you put in to evaluate our manuscript. Your feedback has been invaluable to our work. Please find the revised version of the paper enclosed. In newly enclosed revised version of the paper, the new changes in this second revised version of the manuscript are highlighted in yellow color. Track changes show the changes in the second revision of the manuscript.

The following are our responses to your comments. Please note that the referees’ comments are written in green and our responses in black. Also, the by-one-by response to the comments are as follows. 

 The abstract needs to have a smooth flow in terms of the discussions. Don't include pointwise discussions.

Thank you for your comment. To make the abstract have a smooth flow, especially in terms of the discussion, we re-write the abstract, as follows:

Effective logistics management is crucial for the distribution of perishable agricultural products to ensure they reach customers in high-quality condition. This research examines an integrated, multi-echelon supply chain for perishable agricultural goods. The supply chain consists of four stages: supply, processing, storage, and customers. This study investigates the quality-related costs associated with product perishability to maximize supply chain profitability. Key factors considered include the network design, location of processing and distribution centers, the ability to process raw products to minimize post-harvest quality degradation, the option to sell the excess produce to a secondary market due to unpredictable yields, and the decision not to fulfill demand from distant customers where significant quality loss and price drops would be involved, instead diverting those products to the aforementioned secondary market. Quantitative methods and linear mathematical programming are employed to model and validate the proposed supply chain using actual data from a real-world case study on vegetable supply chains. The main contribution of this research is the incorporation of quality costs into the objective function, which allows the supply chain to prioritize meeting nearby customers' demands with minimal quality loss over serving distant customers where high quality loss is unavoidable. Additionally, deploying a faster transportation fleet can significantly improve the overall profitability of the perishable product supply chain

 Many subsections were included, which prevented the readers from having a smooth reading. Kindly adjust and remove the subsection numbers.

Thank you for your comment. The subsection numbers are made to only one and second level. The third level of the subsection numbers is removed, as it is apparent in the manuscript. Please check it in the new one.

 Data availability references and tables/data-based in-depth information are not given.

Thank you for your comment. The more precise data availability references and data-based in-depth information are stated in the case study section, as follows:

In this section, we implement the proposed model in an Iranian raw and processed vegetable products’ company, the Razian Company, as a case study. 

The case study used a four-echelon SCND, and materials were supplied, processed, and stored (echelons 1-3) in the firm area (origin) while the last-level centers were located all over the country; in addition, a center was established as a second market to collect the in-excess products, as shown in Fig. 1. The mentioned lateral market imposes no costs on the supply chain because it is closest to farms, and customers pay the transportation costs. 

At first, the firm seasonally provided the vegetables from the suppliers. Suppliers were specified and contracted in advance in fertilized source centers (i=4) of selected vegetables (n=3). The farm centers were, in Kaboudrahang, Razan, Nahavand, and Malayer, and the vegetable products were Yarrow, Borage flower, and Melisa. Secondly, the firm used the related processing on vegetables, or the products remained raw. There are potential processing center (j=5) candidates in the case study. Thirdly, the firm stored the products in the storage centers for packaging. There are potential storage center (k=5) candidates in the case study. The five potential processing and storage center candidates were Kaboudrahang, Razan, Nahavand, Malayer, and Asadabad. Finally, the firm delivered the demanded products to the customer centers. The customers were trade representatives of each province all over the country (l=30). Due to the importance of the case study data for the application of the presented model, some were obtained from the enterprise resource planning (ERP) of Razian company (https://razian.co/). In addition, data on fixed and variable costs of different transportation modes were obtained from the recent case study research done in Iran [39]. Data on the price of different raw and processed vegetable products were gathered from the statistics of the Ministry of Agriculture [56]. Details of the most critical data of the case study are presented in Table S1 in the Appendix. 

Of course, the Appendix is a separate file attached.

 The paper needs clearer insights into future trends and the ability of researchers to pursue them based on existing research. More detailed discussions on future research directions, trends, challenges, and opportunities are needed.

Thank you for your comment. As you mentioned in this valuable comment, we re-write the limitations and future research section, with a more detailed discussion on the future research directions and opportunity to do valuable research in the future for those whom are passionate about the field of agricultural products supply chain design, as follows:

Our framework is limited in some respects. With that said, this modeling limitations serve as a platform for extending it in future researches. One primary limitation of the presented model is that it does not consider the uncertainty in the amount of customers’ demand. Therefore, the proposed model does not work for the problem in uncertain conditions. Also, the proposed model in this research has been solved by the exact-type solving method of mathematical programming, which is proper for solving the small size of problems such as the studied case. Considering the limitations above, using mathematical models by uncertainty considerations in the supply chain parameters and applying meta-heuristic methods to solve medium and large-sized problems are suggested in the future research. From the managerial perspective, the presented research works by the assumption of that upstream suppliers, freight transportation, processing centers, and storage facilities are integrated and it needs to build alignment between their organizations to deploy the solutions proposed by the output of the proposed framework. For these efforts to be successful, for future research, it is suggested to study how to cooperate all parties involved in the supply chain, and design the coordination infrastructure in the supply chain to yield the positive effects of proposed supply chain network design, in practice. 

Reviewer 3:

Dear reviewer,

Thank you for the time you put into evaluating our manuscript. We are grateful to you for your positive feedback to our work.

---

## [Editor Report · Decision Letter 3]

19 Apr 2024

A Multi-level Multi-Product Supply Chain Network Design of Vegetables Products Considering Costs of Quality: A Case Study

PONE-D-23-27530R3

Dear Dr. sareh khazaeli,

We’re pleased to inform you that your manuscript has been judged scientifically suitable for publication and will be formally accepted for publication once it meets all outstanding technical requirements.

Kind regards,

Md. Monirul Islam, PhD

Academic Editor

PLOS ONE

Additional Editor Comments (optional):

Dear author

Well done. Best of luck.
---

## [Editor Report · Acceptance letter]

22 May 2024

PONE-D-23-27530R3 

PLOS ONE

Dear Dr. khazaeli, 

I'm pleased to inform you that your manuscript has been deemed suitable for publication in PLOS ONE. Congratulations! Your manuscript is now being handed over to our production team.

Kind regards, 

on behalf of

Dr. Md. Monirul Islam 

Academic Editor

PLOS ONE